# The MIPAS/Envisat climatology (2002–2012) of polar stratospheric cloud (PSC) volume density profiles

Michael Höpfner[1], Terry Deshler[2], Michael Pitts[3], Lamont Poole[4], Reinhold Spang[5], Gabriele Stiller[1], and Thomas von Clarmann[1]

[1]Institute of Meteorology and Climate Research, Karlsruhe Institute of Technology, Karlsruhe, Germany
[2]Department of Atmospheric Science, University of Wyoming, Laramie, Wyoming, USA
[3]NASA Langley Research Center, Hampton, Virginia, USA
[4]Science Systems and Applications, Incorporated, Hampton, Virginia, USA
[5]Institut für Energie und Klimaforschung, Stratosphäre, IEK-7, Forschungszentrum Jülich, Jülich, Germany

**Correspondence:** M. Höpfner (michael.hoepfner@kit.edu)

**Abstract.** A global data set of vertical profiles of polar stratospheric cloud (PSC) volume density has been derived from Michelson Interferometer for Passive Atmospheric Sounding (MIPAS) space-borne infrared limb measurements between 2002 and 2012. To develop a well characterized and efficient retrieval scheme, systematic tests based on limb-radiance simulations for PSCs from in-situ balloon observations have been performed . The finally selected wavenumber range was $831–832.5 \, \text{cm}^{-1}$.

Optical constants of nitric acid trihydrate (NAT) have been used to derive maximum and minimum profiles of volume density which are compatible with MIPAS observations under the assumption of small, non-scattering and larger, scattering PSC particles. These max/min profiles deviate from their mean value at each altitude by about 40-45%, which is attributed as the maximum systematic error of the retrieval. Further, the retrieved volume density profiles are characterized by a random error due to instrumental noise of $0.02–0.05 \, \mu\text{m}^3\text{cm}^{-3}$, a detection limit of about $0.1–0.2 \, \mu\text{m}^3\text{cm}^{-3}$ and a vertical resolution of around $3 \, \text{km}$. Comparisons with co-incident observations by the Cloud-Aerosol Lidar with Orthogonal Polarization (CALIOP) on the CALIPSO (Cloud-Aerosol Lidar and Infrared Pathfinder Satellite Observations) satellite showed good agreement regarding the vertical profile shape. Quantitatively, in the case of supercooled ternary solution (STS) PSCs, the CALIOP dataset fits to the MIPAS retrievals obtained under the assumptions of small particles. Unlike for STS and NAT, in the case of ice PSCs the MIPAS retrievals are limited by the clouds becoming optically thick in the limb-direction. In these cases, the MIPAS volume densities represent lower limits. Among other interesting features, this climatology helps to study quantitatively the on-set of PSC formation very near to the South Pole and the large variability of the PSC volume densities between different Arctic stratospheric winters.

## 1 Introduction

Polar stratospheric clouds (PSCs) form when temperatures fall below about $195 \, \text{K}$ in the Arctic and Antarctic stratosphere during wintertime. They consist either of $HNO_3$/ $H_2SO_4$/ $H_2O$ supercooled ternary solution droplets (STS), of nitric acid trihydrate crystals (NAT) or of $H_2O$ ice particles (Peter, 1997; Solomon, 1999; Peter and Grooß, 2012) . PSCs play a crucial

role in the depletion of ozone by providing the surface and volume for heterogeneous reactions. Through these reactions the chlorine reservoir gases HCl and $ClONO_2$ are converted into ozone depleting forms of chlorine. Beside this primary process, PSCs may foster ozone depletion by sedimentation of large NAT particles. This leads to depletion of stratospheric air of $HNO_3$ (denitrification) by which the reformation of $ClONO_2$ is delayed (Fahey et al., 1990; Solomon, 1999).

Though the relevant processes of PSCs in ozone destruction are well accepted since many years, various aspects are still under discussion. For example, the processes by which NAT particles nucleate are not entirely understood (Peter and Grooß, 2012; Hoyle et al., 2013). Further, our knowledge on size and shape of large NAT particles is incomplete, but largely relevant to the efficiency of denitrification (Woiwode et al., 2014, 2016). Also knowledge gaps related to the formation mechanisms of ice PSCs have been discussed recently (Engel et al., 2013) and even the pathways controlling chlorine activation are still under

investigation (Drdla and Müller, 2012; Wegner et al., 2012; Wohltmann et al., 2013; Kirner et al., 2015; Wegner et al., 2016; Nakajima et al., 2016). In the future, PSCs and related processes leading to ozone depletion will be influenced by the changing stratospheric environment through the changes in temperature, trace gas concentrations ($H_2O$ and $HNO_3$) and small-scale processes such as mountain waves (e.g. Orr et al., 2015). Thus, for a more accurate prediction of future polar ozone depletion a better understanding of processes involving PSCs is desirable.

In the beginning of the 21st century, the capabilities for observing PSCs and related trace gases covering the wintertime polar stratosphere have improved significantly, mainly due to three space-borne missions: the Cloud-Aerosol Lidar with Orthogonal Polarization (CALIOP) on the CALIPSO (Cloud-Aerosol Lidar and Infrared Pathfinder Satellite Observations) satellite (since 2006) (Winker et al., 2009), the Microwave Limb Sounder (MLS) on Aura (since 2004) (Waters et al., 2006), and the Michelson Interferometer for Passive Atmospheric Sounding (MIPAS) on Envisat (2002–2012) (Fischer et al., 2008).

Due to the insensitivity of the microwave spectral region to PSCs, MLS observations have been used for the characterization of the gas-phase stratospheric composition - thereby allowing to derive indirect characterization of PSCs in combination with the direct observations by CALIOP (Lambert et al., 2012, 2016). Flying within the A-train in formation with MLS, CALIOP provides cross-sections of PSC backscatter and depolarization with hitherto unprecedented spatial resolution in the vertical and along track (Pitts et al., 2007, 2009, 2011, 2013).

In the case of MIPAS, investigations have concentrated on occurrence of PSCs, cloud top altitude, and their composition derived from specific features in the infrared limb spectra (Spang et al., 2005; Höpfner et al., 2006a; Spang et al., 2018). In these studies, altitude-dependent parameters, like PSC existence and composition, are derived based on each single limb-view separately, i.e. without consideration of the fact that each raypath of the observation intersects multiple altitude levels, leading to an intertwined retrieval problem for a complete limb sequence. In the present work we tackle this problem by adopting a

complete altitude-resolved inversion of all views of one limb-scan simultaneously. This means that, like in the case of standard trace gas retrievals, a global fit approach is used to derive altitude profiles of PSC volume densities (e.g. Höpfner et al., 2006b).

Beyond the PSC existence and composition, which is already available from MIPAS (Spang et al., 2018), volume density is an independent quantitative parameter which can be used for validation and analysis of atmospheric model results. For example, by comparison with MIPAS data on volume density, Khosrawi et al. (2018) could show, that their global model simulates PSC

existence well but underestimates strongly the PSC mass which might influence vertical redistribution of $HNO_3$.

In the present work we report on the first complete limb retrievals of altitude profiles of PSC volume density over the entire period of MIPAS observations from 2002–2012.

## 2 MIPAS on Envisat

MIPAS was operated on ESA's sun-synchronous polar orbiter Envisat from June 2002 until April 2012. The limb-scanning instrument analyzed the mid-infrared radiation between 4.1 and 14.6 µm emitted by atmospheric trace gases as well as emitted and scattered by clouds and aerosols (ESA, 2000; Fischer et al., 2008). Two major time periods with different standard operation modes can be distinguished: from June 2002 to April 2004 a spectral resolution of $0.025\,\mathrm{cm}^{-1}$ in combination with a vertical sampling of $3\,\mathrm{km}$ tangent point distance in the altitude region of PSCs combined with an along-track horizontal sampling of $530\,\mathrm{km}$ was in operation. Then, from January 2005 until April 2012, the spectral resolution was changed to $0.0625\,\mathrm{cm}^{-1}$ while the spatial sampling was improved to $1.5\,\mathrm{km}$ up to $22\,\mathrm{km}$ altitude and to $2\,\mathrm{km}$ up to $32\,\mathrm{km}$ altitude. An improved along-track sampling of $400\,\mathrm{km}$ was achieved. With these two different sampling patterns, MIPAS provided about 1000 limb-profiles per day during the first period increasing to 1400 scans per day in the second phase. In regions poleward of $60°$ latitude about 170 and 240 profiles per day have been obtained during each period, respectively. Due to the possibility to turn the pointing azimuth of the limb-sounder away from the orbital plane, the MIPAS tangent points cover the polar area nearly up to the poles (from $87.5°$S to $89.3°$N), while nadir pointing instruments, like CALIOP, reach maximum polar latitudes of about $82°$. The following retrievals are based on MIPAS level-1b calibrated radiances of data version 5.02/5.06 as provided by ESA (Nett et al., 2002).

## 3 Retrieval

Günther et al. (2018) have described the retrieval of stratospheric sulfate aerosol volume density profiles from MIPAS observations. Their analysis relies on the assumption of sulfuric acid as the major component of stratospheric aerosol. The particle sizes of this type of aerosol can be assumed to be less than one micrometer in radius, which simplifies radiative transfer simulations considerably. For absorbing aerosols small compared to the wavelength, scattering effects can be neglected, and the radiance only depends on the volume emission of the particles (e.g. Bohren and Huffman, 2008, p. 130).

In case of PSCs, the retrieval of volume density profiles is complicated by the fact that PSCs consist of particles of different composition belonging to different size classes. STS particles are generally smaller than one micrometer, and, thus fall in the same category as sulfate aerosols regarding their emission of thermal infrared radiation. On the other hand, NAT and ice particles are often larger than $1\,\mu\mathrm{m}$ and, thus, scatter radiation from the troposphere below into the instrument's line-of-sight in addition to their thermal emission (Höpfner et al., 2002; Höpfner, 2004; Höpfner et al., 2006b). Further, in many cases not only PSCs of a single homogeneous composition are present within the field-of-view of the limb sounder. As, e.g., observed by CALIOP, a mixture of particles with different composition in the same airmass or several thin layers of a single composition may be present in the observed volume (Höpfner et al., 2009; Pitts et al., 2018). This leads to a complex retrieval problem

since the measured spectra are not only influenced by the radiative effects of particles with different composition, but also by the different sizes of the different PSC particle types (Höpfner et al., 2006b).

To simplify this inversion problem in a way that it becomes applicable for the full MIPAS dataset, we have adopted the following approach. In a first step we have produced a test dataset by simulating MIPAS limb radiances of PSCs including scattering of tropospheric radiation for different tropospheric cloud scenarios. These synthetic observations provide the basis for simulations to optimize the retrieval settings with respect to wavelength range and refractive index set to derive PSC volume density profiles. This parameter set is subsequently used to process the entire MIPAS dataset on basis of which comparisons with measurements by CALIOP has been performed.

In the following these different steps are presented in more detail.

## 3.1  Radiative transfer modelling

The radiative transfer model KOPRA (Karlsruhe Optimized and Precise Radiative transfer Algorithm) (Stiller, 2000) is suited for analysis of spectrally high resolved remote sensing observations of any observational geometry (Hase and Höpfner, 1999). Based on line-by-line calculations, the model takes into account all relevant physical effects in the lower and middle atmosphere (Kuntz, 1997; Kuntz and Höpfner, 1999). Radiative transfer in presence of particles is handled with a single scattering approach and the embedded Mie model allows the direct input of microphysical particle properties as well as the calculation of Jacobians with respect to these parameters (Höpfner et al., 2002; Höpfner, 2004). KOPRA has been validated extensively regarding gas-phase (Glatthor et al., 1999; Tjemkes et al., 2001; von Clarmann et al., 2002, 2003; Schreier et al., 2018) and aerosol radiative transfer (Höpfner and Emde, 2005).

## 3.2  Retrieval model

KOPRAFIT, a non-linear retrieval environment of KOPRA, is applied as a standard tool for data analysis of infrared limb and nadir observations from satellite, balloon and aircraft (e.g. Wetzel et al., 2015; Höpfner et al., 2009; Keim et al., 2009; Woiwode et al., 2016). The software allows the direct derivation of microphysical properties of particles from radiance spectra (Höpfner et al., 2002; Höpfner et al., 2006b; Höpfner, 2008). It is based on the internal calculation of single scattering properties by the embedded Mie-model using altitude dependent lognormal size distribution parameters.

The applied retrieval approach is a constrained non-linear multi-parameter fit of the simulated limb-radiances to the observed spectra. Atmospheric profiles of aerosol parameters are represented by the vector of unknowns, $\boldsymbol{x}$, which is determined in a Newtonian iteration process to account for the nonlinearity of the atmospheric radiative transfer (Rodgers, 2000):

$$\boldsymbol{x}_{i+1} = \boldsymbol{x}_i + (\mathbf{K_i}^T \mathbf{S}_y^{-1} \mathbf{K_i} + \mathbf{R})^{-1} \times (\mathbf{K_i}^T \mathbf{S}_y^{-1} (\boldsymbol{y}_{\mathrm{meas}} - \boldsymbol{y}(\boldsymbol{x}_i)) - \mathbf{R}(\boldsymbol{x}_i - \boldsymbol{x}_a)). \tag{1}$$

$\boldsymbol{y}_{\mathrm{meas}}$ contains measured spectral radiances of all limb views, and $\mathbf{S}_y$ is the related measurement noise covariance matrix. $\boldsymbol{y}(\boldsymbol{x}_i)$ contains the spectra calculated by the radiative transfer model using the best guess atmospheric state parameters $\boldsymbol{x}_i$ of iteration number $i$. $\mathbf{K_i}$ is the Jacobian matrix, i.e., the partial derivatives $\partial \boldsymbol{y}(\boldsymbol{x}_i)/\partial \boldsymbol{x}_i$. $\mathbf{R}$ is a regularization matrix and $\boldsymbol{x}_a$ the a-priori information.

The retrieval tests below are performed at 500 m spaced vertical grid levels. This implies undersampling by the measurements and, thus, a vertical constraint is needed to avoid retrieval instabilities. For the regularization, $\mathbf{R}$ a first-order smoothing constraint $\mathbf{R} = \gamma \mathbf{L^T L}$ with the altitude-independent regularization parameter $\gamma$. $\mathbf{L}$ is a first order finite differences operator (Tikhonov, 1963).

To determine the degrees of freedom and the vertical resolution of the retrieved altitude profiles, the averaging kernel is analyzed:

$$\mathbf{A} = (\mathbf{K}^T \mathbf{S}_y^{-1} \mathbf{K} + \mathbf{R})^{-1} \mathbf{K}^T \mathbf{S}_y^{-1} \mathbf{K}. \tag{2}$$

From $\mathbf{A}$ the vertical resolution can either be determined in terms of the full width at half maximum of the related column of the averaging kernel matrix or by the grid level distance divided by the diagonal values of $\mathbf{A}$.

## 3.3 Synthetic MIPAS observations based on in-situ data

We have used realistic atmospheric input parameters to simulate MIPAS limb radiances. For profiles of PSC size-distributions, the University of Wyoming database of balloon-borne observations (http://www-das.uwyo.edu/~deshler/Data/Aer_Meas_Wy_ read_me.htm) has been chosen. These data result from in-situ size-resolved aerosol concentration measurements using balloon-borne aerosol counters. The datasets with PSC observations have been collected during flights from McMurdo and Kiruna (Deshler et al., 2003). This is by far the largest set of consistent in-situ particle size distribution observations of PSCs. It determines the number (128) of independently simulated MIPAS limb-scans of our test dataset. Beside the original data which consist of the particle number densities within distinct size classes, the database provides one- and two-modal lognormal size distribution parameters (total number concentration, median radius and geometric standard deviation) calculated from size distribution fits to the aerosol measurements. These parameters can directly be applied as input data for the reference radiative transfer calculations with the KOPRA forward model.

Figure 1 shows eight examples from the balloon database. The distinction between STS, NAT and ice has been made on basis of size and volume density. The first mode with median radii much smaller than 1 µm has been attributed to STS, indicated by dotted profiles of median radius and volume density. The second size mode has been assigned to NAT with the exception when the volume densities of the second mode reach values larger than $10 \, \mu m^3 cm^{-3}$. In that case ice was used instead of NAT over the whole vertical range of the profile.

In addition to the altitude-dependent particle size parameters, the balloon database provides the supplementary meteorological dataset, especially for pressure and temperature. We have compared these to profiles derived from ERA-Interim re-analysis data (http://apps.ecmwf.int/datasets/data/interim-full-daily/levtype=pl/) interpolated to the position of the balloon. In general, the difference between ERA-Iterim and the in-situ observed temperatures are about 1–2 K. The standard deviations between the single profiles amount to 2–3 K. For the reference radiative transfer calculations we have decided to use the pressure/temperature profiles derived from ERA-Interim since these provide consistent pressures and temperatures over a large altitude range suited for simulation of limb radiances. Furthermore, also the surface temperatures for each individual balloon profile are obtained from ERA-Interim analyses ("2 m temperatures") for the site of the balloon launch. Trace gas profiles

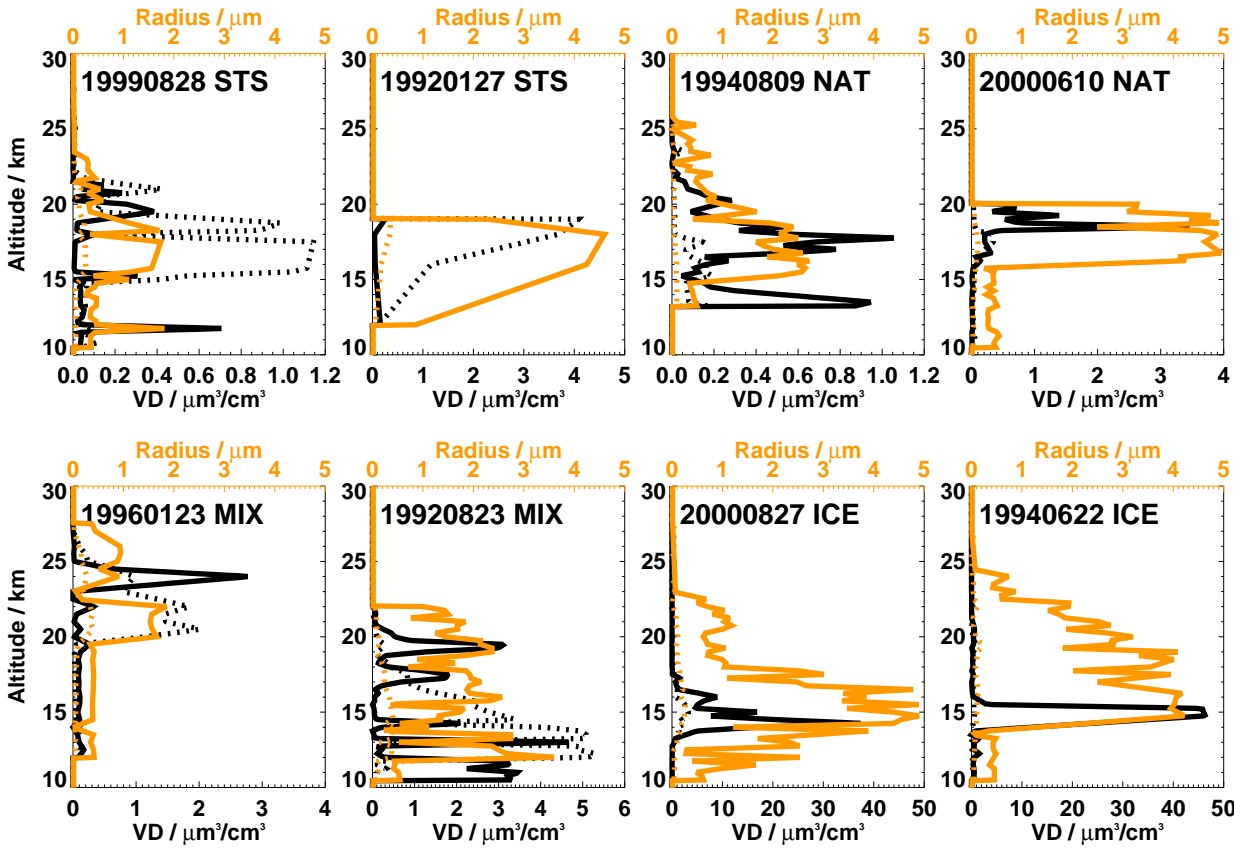

**Figure 1.** Example profiles from the in-situ balloon database on PSCs used as input for the radiative transfer model. The database contains parameters of bi-modal log-normal distributions derived from the particle counter measurements. Here the median radius (top orange axis) and the total particle volume density (VD, bottom black axis) of each mode are shown. Dotted lines indicate the first mode with smaller particles and solid lines the second mode. The title indicates the date of the balloon observation and the predominant composition of the PSCs (MIX is a mixture of similar volume densities of STS and NAT).

are obtained from polar winter standard atmospheres (Remedios et al., 2007). To account for a variability of the the actual tropospheric cloud situation during the balloon observations, we have performed the reference calculations for the following situations: (a) cloud-free, (b) an (in nadir direction) optically thick cloud at 6 km and, (c) an optically thick cloud at 8 km altitude.

Höpfner et al. (2006b) investigated MIPAS observations of PSCs of single-composition for the best fit to radiative transfer simulations. It could be shown, that over large wavenumber regions, the refractive indices of $\beta$-NAT by Biermann (1998) fitted best the observed spectra of NAT PSCs, those by Biermann et al. (2000) of STS PSCs , and ice PSC spectra could be modeled well by use of data from Toon et al. (1994). For STS, the altitude dependent refractive indices have been determined by the mixing rule provided by Biermann (1998). As input for the mixing rule, the STS particle composition has been calculated from thermodynamic equilibrium (Carslaw et al., 1994) based on the actual temperature, standard winter $HNO_3$ and $H_2O$ mixing-ratio profiles, and 0.3 ppbv of $H_2SO_4$.

## 3.4 Development of the retrieval configuration

Using the PSC limb radiance calculations described in the previous section, retrieval simulations for aerosol volume density have been performed using in total twelve combinations of (a) composition, and, (b) spectral windows. For composition, refractive indices for (1) STS, (2) NAT, and (3) ice have been applied (see above). For spectral windows, wavenumbers around (1) 831–832.5 $cm^{-1}$, (2) 956.5–957.5 $cm^{-1}$, (3) both of (1) and (2) combined, and, (4) 1226.5–1227.5 $cm^{-1}$ have been used. These infrared windows are characterized by the smallest interference of trace gases within the spectral bands of MIPAS. The following trace gases have been considered within the radiative transfer simulations: $H_2O$, $CO_2$, $O_3$, $N_2O$, $CH_4$, $HNO_3$, $C_2H_6$, $CFC-11$, $CFC-22$. The retrievals have been performed assuming small particles for which only the volume absorption and emission influences the limb radiances.

Fig. 2 shows an overview of the differences between the reference volume density profiles and the retrieved ones. For these overview tables, the in-situ profiles have roughly been classified with respect to their major composition as STS, NAT, ICE, and MIXed cases (vertical axis). The twelve retrieval test cases described above are distinguished along the horizontal axis.

The left part of Fig. 2 refers to the altitude range between 16 and 20 km and the right part to 20–24 km. It is obvious that different retrieval settings show a different performance depending on the reference case. When inspecting the columns for those assumptions resulting in the best overall performance, either NAT in window 1 ("refra_NAT_mw_1") or ICE in window 1 ("refra_ICE_mw_1") appear superior compared to the rest of combinations of composition and spectral window.

Based on these tests we have chosen the combination of window 1 together with NAT optical constants as our baseline retrieval configuration. We prefer the use of NAT spectroscopic data since, compared to ice, these kinds of PSCs (as well as mixtures of STS and NAT) are more often detected than ice. Especially in the northern polar winter, ice-PSCs are very rare due to the higher temperature levels. But also over Antarctica, ice-PSCs are not as frequent as NAT/STS, especially during the inital phases of the winter and at lower latitudes (Spang et al., 2018).

Figure 3 presents the results obtained on the basis of the selected retrieval configuration for the examples shown in Fig. 1. The retrieved profiles (blue lines) for the three different situations of tropospheric cloud cover are compared to the balloon profile

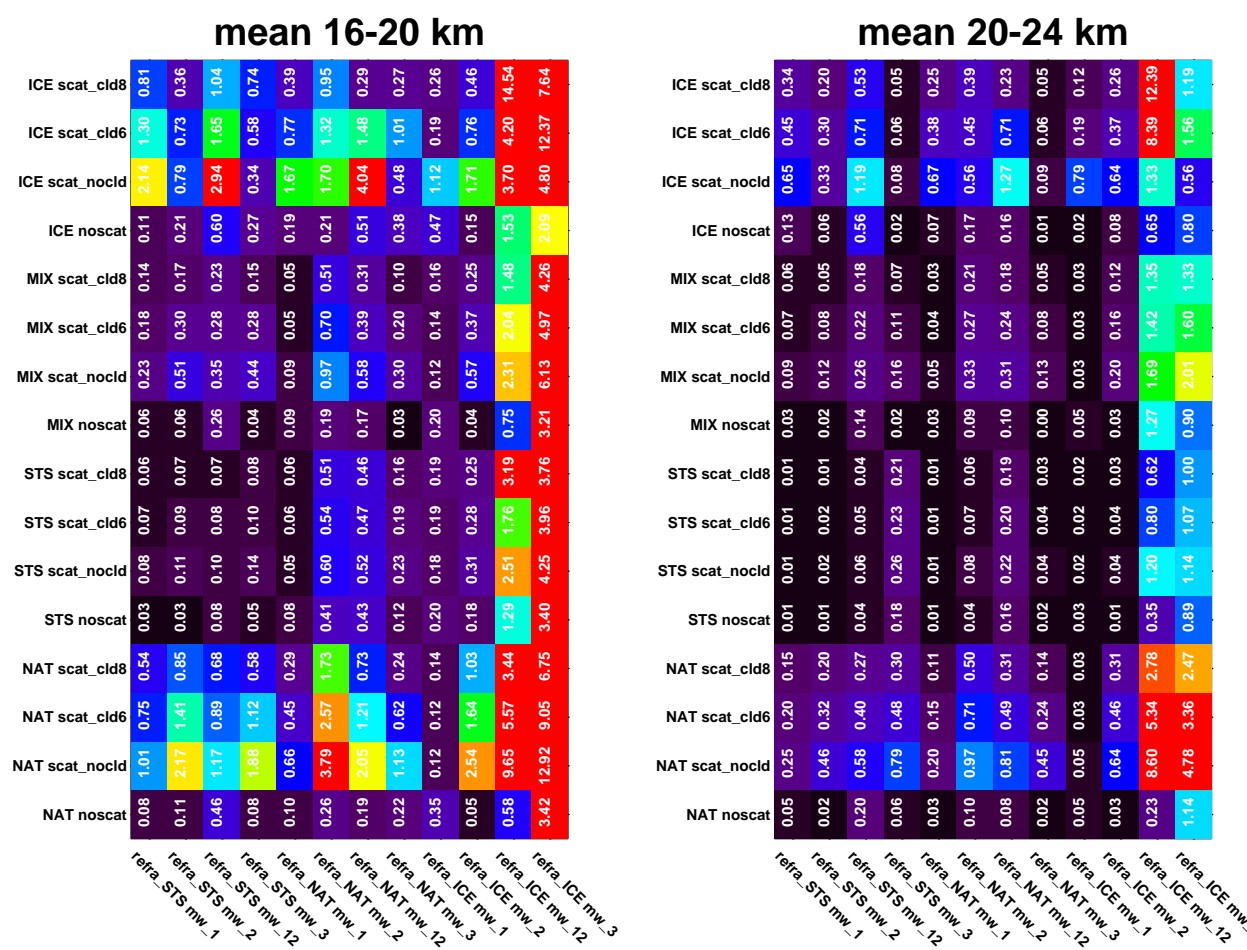

**Figure 2.** Results of the PSC retrieval sensitivity study. The numbers and the color code of each box indicate the absolute values of the mean differences in volume density ($\mu m^3 cm^{-3}$) between the retrieval result and the in-situ profiles within the altitude regions 16–20 km (left) and 20–24 km (right). Each column contains the results for a specific retrieval assumption on spectral window (831–832.5 $cm^{-1}$ ("mw_1"), 956.5–957.5 $cm^{-1}$ ("mw_2"), both ("mw_12"), and 1226.5–1227.5 $cm^{-1}$ ("mw_3")) and used refractive index ("refra_STS, refra_NAT, refra_ICE"). Each row shows the residuals for a specific assumption of the tropospheric cloud scene ("nocld, cld6, cld8") for a sub-set of in-situ profiles of PSCs consisting mainly of ice, STS, NAT and NAT/STS mixtures ("MIX").

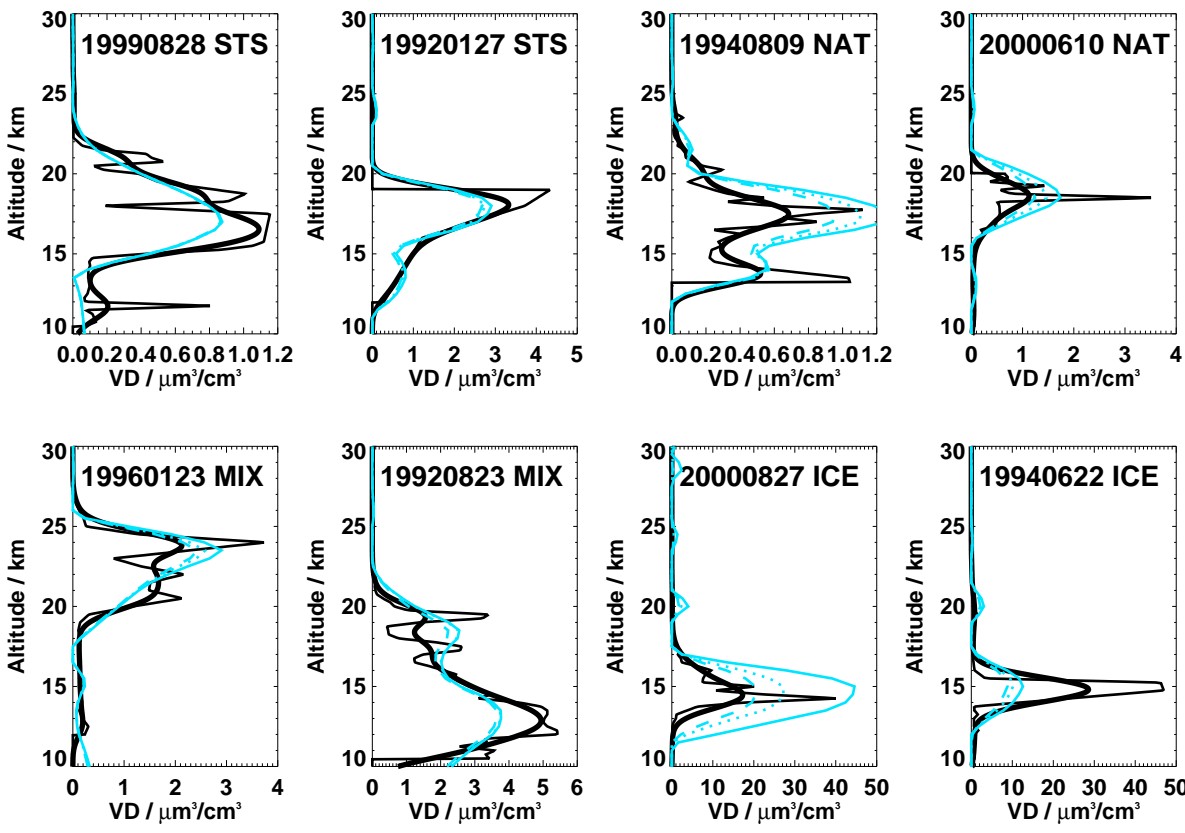

**Figure 3.** Total volume densities of the balloon profiles shown in Fig. 1 without (thin black lines) and with convolution by the MIPAS averaging kernel (bold black lines). The light blue curves are the retrieval results for the chosen retrieval configuration with cloud-free troposphere (solid), with a tropospheric cloud with 6 km cloud-top altitude (dotted) and a cloud at 8 km (dashed). The title indicates the date of the balloon observation and the predominant composition of the PSC (MIX is a mixture of similar volume densities of STS and NAT).

of total PSC volume density in original (thin black) and transferred to the vertical resolution of MIPAS (bold black). Where small STS particles dominate the total volume (first two panels), the retrieval results compare within about 0.2–0.4 $\mu$m$^3$cm$^{-3}$ with the reference. In case of NAT as the predominant composition, the volume densities are generally overestimated, since scattering is neglected in the retrieval. It can well be observed that the less scattering contributes from the troposphere, which is the case for a cold tropospheric cloud at 8 km, the better the result fits the reference. For mixed profiles of NAT and STS an over-, but also some underestimation is possible. In case ice PSCs dominate the volume, the retrieval leads to overestimation as in case of NAT. However, for larger volume densities, an underestimation by a factor of about 2–3 is observed. This is explained by saturation effects, because the limb views become optically thick at volume densities of a few tens of $\mu$m$^3$cm$^{-3}$.

In summary, in this section we have described the use of simulated limb-measurements, using real in-situ observations as input to develop a fast retrieval configuration based on one spectral window, one set of optical constants and the assumption of volume emission by small particles.

## 4 Characterization of the retrieval and comparison with CALIOP

To characterize in greater depth the method for retrieval of PSC volume density as introduced above, we have applied it to real MIPAS PSC observations during the Antarctic winter of 2009. To be able to judge these retrievals against independent observations, co-incident measurement locations with the CALIPSO lidar instrument have been selected within a circle of 200 km distance and a maximum time difference of 2 h from each MIPAS limb-scan.

From the CALIOP PSC observations, new data products on surface area density and volume density have become available recently (Pitts et al., 2018). Their estimated uncertainties of volume density derived in case of STS PSCs are in the range of 0.05–1.0 $\mu$m$^3$cm$^{-3}$. For NAT mixtures and ice PSCs, the CALIOP volume density values are mostly lower limits and can be underestimated by factors of 10 and up to 30 for NAT and ice PSCs, respectively. Thus, additionally to the vertical location of the PSCs, it has been possible to perform quantitative comparisons of the volume density data products between CALIOP and MIPAS.

In Figures 4 and 5 we present examples of comparisons between volume density profiles retrieved from single MIPAS limb-scans and co-incident profiles derived from CALIOP. The altitude dependent characterization of the related MIPAS retrievals regarding spectral noise error (black) and vertical resolution (red) are given by the panels in the third Figure column. The vertical resolution, as derived from the diagonal of the averaging kernel, varies around 2.5–3.5 km in nearly all cases. Measurement noise causes a retrieval error of about 0.02–0.08 $\mu$m$^3$cm$^{-3}$.

As indicated by the PSC-type information of the CALIOP lidar (5th column in Fig. 4 and 5), we have selected cases with STS as the predominant composition in the top two rows of Fig. 4, followed by mixtures between STS, NAT-containing particles (rows 3–4 and Fig. 5). For the last three cases in Fig. 5, CALIOP indicates ice with increasing fraction. All these examples show that the vertical extent and the qualitative shape of the vertical profiles of both instruments fit well. Even secondary PSC layers above the main peak do show up in both datasets (first and second row of Fig. 5) at about 25 km altitude).

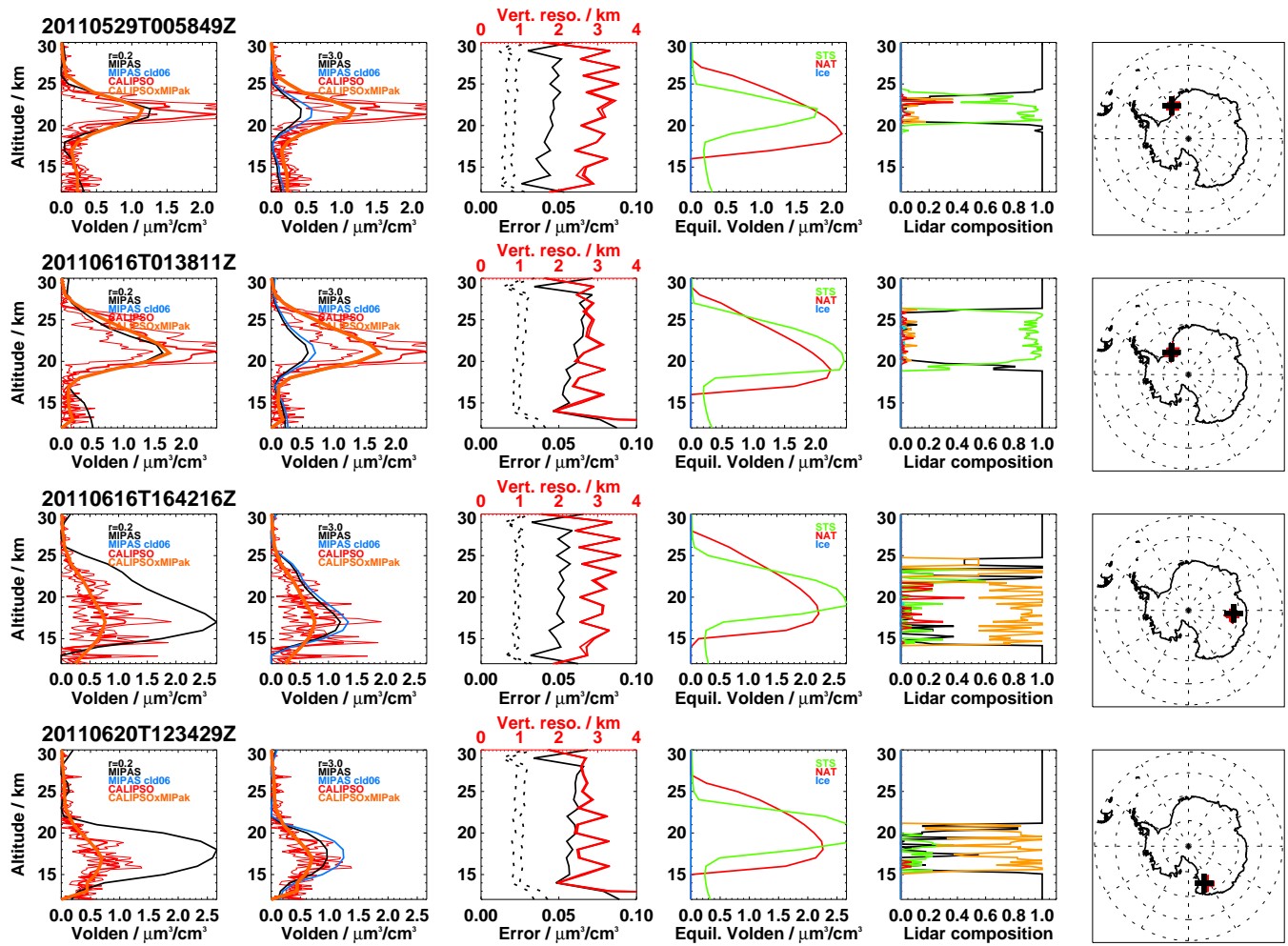

**Figure 4.** Examples of volume density profiles derived from CALIOP and MIPAS for observations within 200 km and 2 h during the Antarctic winter in 2011. First two columns: CALIOP single profiles as thin red curve and their mean as bold red and, multiplied with the MIPAS averaging kernels, as bold orange line. Black profiles in the 1st column, $VD_{max}$: MIPAS retrievals assuming the small particle limit, and in the 2nd column, $VD_{min}$: MIPAS retrievals for a median radius of 3 μm with clear troposphere below. Blue lines are the MIPAS retrievals as $VD_{min}$ but with a tropospheric cloud at 6 km altitude. Third column, red lines: MIPAS vertical resolution, black: estimated retrieval noise error for the results in the first (solid) and the 2nd column (dotted). The fourth column contains the PSC volume densities in case of thermodynamic equilibrium (see text). Fifth column: altitude profiles of the relative composition derived from all co-incident CALIOP data: green is STS, red/orange are NAT-containing PSCs and blue denote ice particles. Right column: the black crosses show the geolocation of the MIPAS limb-scans and the thinner red crosses, the co-incident CALIOP observations.

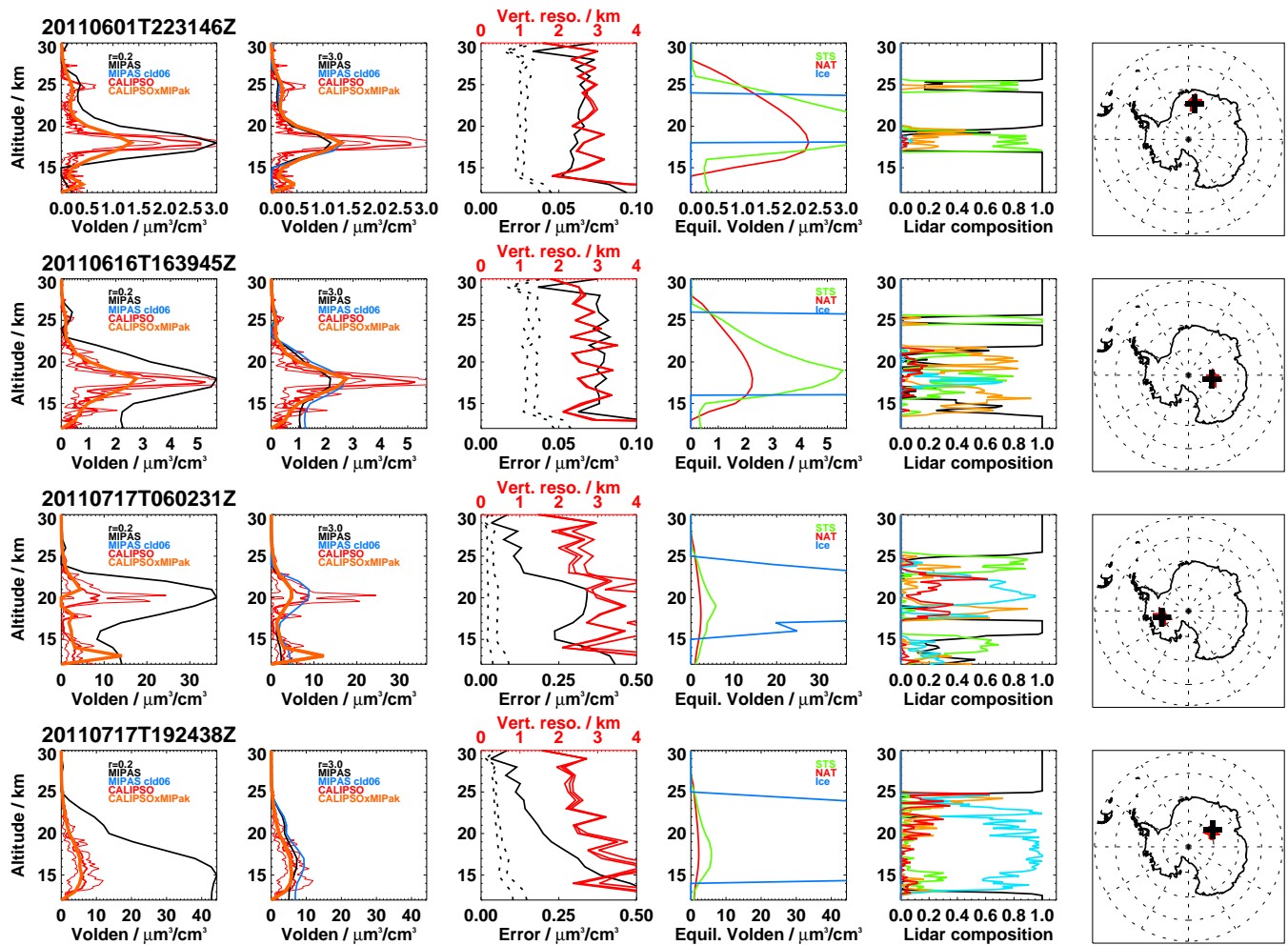

**Figure 5.** (Fig. 4 continued)

In the first two columns of Fig. 4 and Fig. 5 we demonstrate the effect of different assumptions on particle size on the MIPAS retrievals. The first column contains MIPAS volume density profiles retrieved under the assumption of small particles (black, as of now called $VD_{max}$) together with co-incident single CALIOP profiles (thin red), their mean (thick red) as well as the mean lidar profile multiplied with the MIPAS averaging kernel (orange). In the second column, the same lidar profiles

are compared to the MIPAS retrievals for a particle radius of $3\,\mu m$ without (black, called $VD_{min}$ in the following) and with tropospheric clouds at $6\,km$ altitude (blue). Test calculations have shown that particles of about $3\,\mu m$ radius have the strongest contribution of scattered radiation in limb-direction. In consequence, less particle volume is needed to account for the observed limb radiances. This is clearly indicated by the retrieved $VD_{min}$ profiles in the second column which are by about a factor of 2–3 smaller than for the non-scattering assumption. Furthermore, in case of scattering particles, the retrievals depend on the

tropospheric situation below the tangent point. This is quantified by introducing an opaque cloud at $6\,km$ altitude. As can be derived by comparing the blue and black ($VD_{min}$) profiles in the second column of Fig. 4 and Fig. 5, the resulting difference amounts to about 20% larger volume densities when a tropospheric cloud is present, due to the reduced radiation from the scene below the tangent points.

When STS PSCs are predominantly present (first two rows), retrieved $VD_{max}$ values typically fit much better to CALIOP

than those assuming larger particles. This is in agreement with the expectations that STS PSCs consist of particles smaller than $1\,\mu m$ radius and that these volume densities can be well derived from CALIOP. For mixtures of STS and NAT PSCs, the MIPAS small-particle retrievals, $VD_{max}$, in general lead to much larger volume densities compared to CALIOP. However, due to the uncertainties connected to the Lidar volume densities, we cannot unambiguously judge between $VD_{max}$ and $VD_{min}$ when NAT is present.

If there are ice PSCs, it is also difficult to judge on the retrieval quality. First, the Lidar derived volume densities in presence of ice are very uncertain (Pitts et al., 2018). Second, the problems of the limb-retrievals are increased by the fact that an ice-PSC may become optically thick in limb-direction and only a lower limit will be derived by our retrieval. For these cases MIPAS retrievals between the large and the small limit differ by even factors of more than 4. Despite these difficulties in case of ice, it is generally easy to distinguish the ice-cases from the MIPAS observations when $VD_{max}$ shows values of more than about

$10\,\mu m^3 cm^{-3}$ or by the classification of ice by MIPAS PSC classification algorithms (Spang et al., 2018).

Retrieved profiles of particle volume densities can be compared to the volume, solid or liquid PSC phases can reach under thermodynamic equilibrium conditions (Hanson and Mauersberger, 1988; Carslaw et al., 1994). We have calculated these profiles using temperatures from ECMWF, standard polar winter concentration profiles of $HNO_3$ and $H_2O$ (Remedios et al., 2007) and $0.3\,ppbv$ of $H_2SO_4$. When comparing to these equilibrium calculations in the fourth column of Fig. 4 and Fig. 5,

one recognizes that the retrievals of STS and NAT volume densities are generally in the same range of values. However, there are some significant differences when the variation with altitude is taken into account. Here, both, MIPAS retrievals and the CALIOP dataset often indicate much smaller values of volume density compared to the calculations under the assumption of thermodynamic equilibrium. In those cases where ice may be present according to the equilibrium simulations, there is either only little or no ice indicated by CALIOP and the MIPAS retrievals generally show lower volume densities as discussed above.

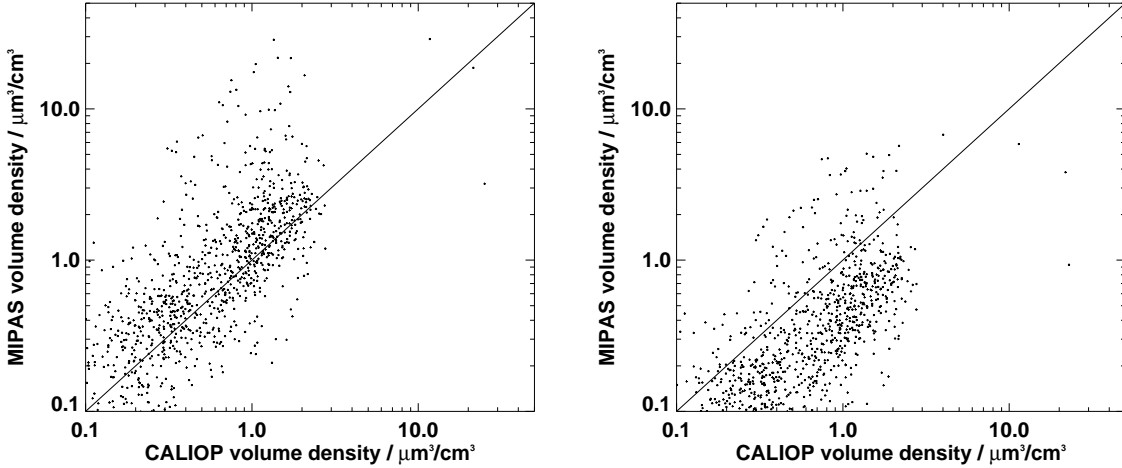

**Figure 6.** Values of PSC volume density retrieved by MIPAS versus co-incident values of CALIOP for cases where the Lidar indicates more than 50% of STS and less than 5% of ice sightings. Left: MIPAS retrieval for small particles ($VD_{max}$), right: MIPAS retrieval for large particles and cloud-free troposphere ($VD_{min}$).

A comparison between volume densities by MIPAS and CALIOP for the whole Antarctic winter is shown in Fig. 6 . We have selected only those co-incident profiles where the lidar indicates a composition of more than 50% of STS and less than 5% of ice. The left part of the Figure contains the MIPAS $VD_{max}$ data, whereas on the right, the $VD_{min}$ retrieval results are shown. Obviously, the assumption of small particles leads to a much better comparison with the CALIOP dataset than in case of

large particles where the volume densities are clearly underestimated. However, there are also some data points of the MIPAS small-particle retrieval which deviate from the behavior of the bulk by exhibiting values larger than about 4–5 $\mu m^3 cm^{-3}$ while the lidar shows values less than 2 $\mu m^3 cm^{-3}$ (left panel of Fig. 6). These outliers may be caused by interference of ice particles in the field of view of MIPAS, which are not identified by the lidar.

Compared to the uncertainties caused by the different assumptions on scattering, the instrumental noise error of around

0.05 $\mu m^3 cm^{-3}$ is small. However, it determines the detection limit of PSCs from the MIPAS dataset as about 0.1–0.2 $\mu m^3 cm^{-3}$. Further it has to be emphasized that we assume homogeneous PSC layers in horizontal direction. Thus, the retrieved volume density values of horizontally restricted PSCs will be underestimated and one has to regard those as the mean values along the limb-path inside a layer of about 400 km length. Due to the large extent of PSC coverage in the southern hemisphere, we expect this assumption to lead to larger uncertainties in the Arctic stratosphere, since there, PSCs are more often constrained

to a smaller regions of cold temperatures.

A further assumption we have made during the development of the retrieval was the choice of the optical constants. Our baseline was to use only those refractive indices of PSC composition and phase which have already been observed in the atmosphere and shown best compatibility with infrared limb observations, i.e. $\beta$-NAT, STS, and ice (Höpfner et al., 2006b). This may lead to the following uncertainties: (1) the optical constants themselves are not perfect (Ortega et al., 2006; Iannarelli and Rossi, 2015). (2) particles may be present with different phases and composition (e.g. $\alpha$-NAT, $\alpha$-NAD, $\beta$-NAD) as laboratory studies indicate their possible existence under polar stratospheric conditions (Grothe et al., 2008; Stetzer et al., 2006; Möhler et al., 2006). And (3), PSC particle shapes different from spherical ones, as seen in the laboratory (Grothe et al., 2006), can have an effect even at wavelengths in the thermal infrared (Wagner et al., 2005; Woiwode et al., 2016). We have implicitly accounted for those errors by the large variability of optical constants of $\beta$-NAT, STS, and ice during the optimisation of the retrieval baseline configuration. Still, the use of one specific set of refractive indices leads to systematic retrieval errors which strongly depend on the atmospheric scene. A validation of infrared limb observations by in-situ measurements, especially of such cases where solid nitric acid containing particles are present, would be helpful to get a better grip on those uncertainties.

## 5  The global MIPAS dataset

For the retrieval of the entire MIPAS dataset covering nine Arctic and nine Antarctic winters between 2002 and 2012 we have adopted the following strategy. In a first step, time periods and spatial coverage was determined on basis of MIPAS PSC detection methods to cover all PSC sightings. In consequence, PSC volume density retrievals were performed for all MIPAS limb-scans from 90°S to 40°S between May, 10th and November, 1st and from 90°N to 40°N between November, 20th and April, 1st, respectively. These calculations could be performed effectively based on the small particle assumption for all selected limb-scans (about 560,000) since no scattering had to be considered. As described above, the resulting profiles can be viewed as an upper limit of volume density ($VD_{max}$). To determine the lower limit, retrievals including scattering are necessary. However, since those are much more costly, we performed these only for observations where the $VD_{max}$ dataset showed maximum volume densities larger than $0.5\,\mu m^3 cm^{-3}$. Still, this resulted in about 160,000 profiles. To estimate the lower limits of volume density for the remaining ~400,000 locations, a mean linear regression derived from the monthly datasets of available volume density values calculated with and without scattering has been utilized. This is shown as Figures 1–5 in the supplementary material.

As an example for the coverage of single profiles, Fig. 7 shows the retrieved PSC volume densities at $20\,km$ altitude in mid-May 2010. One can clearly observe the on-set of PSCs evolution right in the center close to the South Pole. The appearance and mass of PSC over Antarctica in May can deliver valuable information on the nucleation process of NAT particles, which are relevant for denitrification (e.g. Lambert et al., 2016). For example, in Lambert et al. (2016, Tab.2) the reported on-set date as derived from CALIOP is 22-May while according to MIPAS observations (Fig. 7) first PSCs appear 5–6 days earlier. During several of the years observed by MIPAS, the first PSCs are detected in this region during very similar times. These observations are unique since no other instrument has observed PSCs during their formation so far south (Spang et al., 2018).

Figures 8 and 9 provide an overview over the whole MIPAS dataset of PSC volume density retrievals. Here we show daily $10°$ zonal mean values calculated as the mean of the $VD_{min}$ and $VD_{max}$. The equivalent plots containing the results for $VD_{min}$ and $VD_{max}$ separately are provided as Figures A1, A2, A3, and A4 of the appendix. The difference in PSC coverage between the Arctic and Antarctic wintertime stratosphere is clearly visible as well as the large year-to-year variability in the Arctic. In the plots of the southern hemisphere (bottom two rows of Fig. 8) at altitudes of $28\,km$ and $26\,km$, bands of enhanced values are visible during mid-winter. These often appear as side-lobes in the retrieved profile when optically thick ice clouds are present, as can be observed in co-incident observations of CALIOP and MIPAS (Fig. A5). In comparison, high-altitude PSCs are mostly not confined to a single retrieval level, visible in Fig. A6. The instabilities could be suppressed by increasing the regularization strength, however, at the expense of a deterioration of the vertical resolution. We have, thus, decided not to change the constraint, but to point at these potential outliers. Further, there is a background of values up to about $0.5\,\mu m^3 cm^{-3}$ up to about $15\,km$ altitude throughout most of the years - mainly visible in the northern hemisphere. These values may rather be explained by the influence of tropospheric cirrus, which, e.g. at 12-13 km reach the lower edge of the vertical field-of-view of MIPAS pointing at $14\text{-}15\,km$ tangent height, as well as by an increasing influence of aerosol. For example, the enhanced values during the winters of 2008/09, 2009/10 and 2011/12 can be traced back to the enhanced stratospheric aerosol loading caused by the volcanic eruptions of Kasatochi, Sarychev and Nabro (e.g. Günther et al., 2018).

## 6    Conclusions

We have performed the first quantitative retrievals of particle volume density of PSC observations for the whole MIPAS operational period. This is an extension of the available MIPAS datasets on PSC existence, cloud-top and composition as e.g. published by Spang et al. (2018). The retrieval of volume densities of PSCs is complicated through the fact that the particles of NAT and ice PSCs can be so large that the scattered contribution to the detected radiance cannot be neglected. We have tried to minimize this contribution through a systematic selection of spectral window and optical constants. The retrieval configuration was applied to co-incident observations of MIPAS and the CALIPSO lidar in the Antarctic winter of 2011. In case of STS PSCs, the comparisons showed very good agreement between both instruments in terms of the vertical profiles shape as well as the values of volume density. This can be expected since volume emission can be considered as the predominant source of radiation in case of small STS particles. In case of mixed PSCs, these comparisons also support the MIPAS retrievals in terms of vertical profile shape while a quantitative comparison is difficult due to the uncertainties of both datasets. Still, we could demonstrate that retrievals assuming no-scattering and those including scattering of particles with mean radius of $3\,\mu m$ likely cover the entire range of possible volume densities. Thus, we have decided to distribute a dataset containing both, $VD_{max}$ and $VD_{min}$, to provide the possible range of values which are compatible with the MIPAS PSC observations. We are convinced that the dataset is valuable for the validation and analysis of atmospheric model calculations, as e.g. Khosrawi et al. (2018) have reported large discrepancies between their model runs and the MIPAS PSC volume density dataset for the Arctic winter of 2009/10.

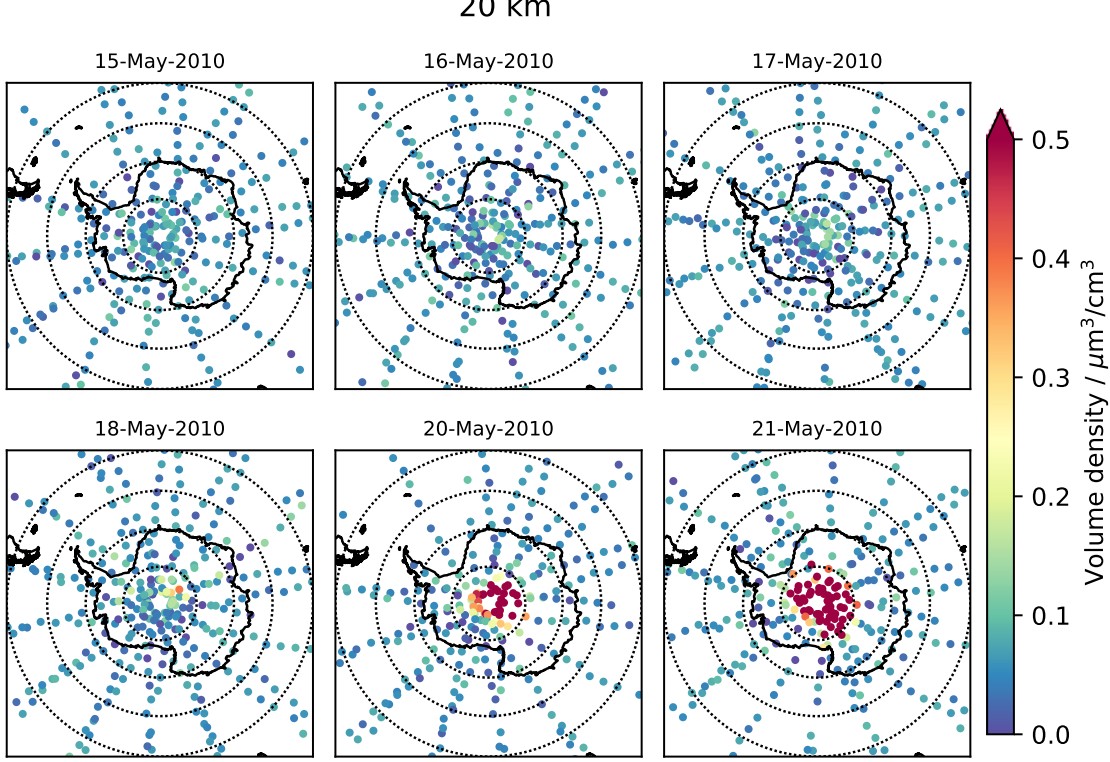

**Figure 7.** Retrieved volume densities at 20 km altitude from single MIPAS limb scans at the beginning of the PSC season over Antarctica in May 2010. The values are averages of the minimum/maximum retrievals as described in the text.

In future, upon availability of larger computer capacities in combination with faster infrared limb scattering models, we believe that this dataset can further be improved. This would allow the exploitation of the broad wavelength coverage of MIPAS which contains information, both, on PSC composition and size. Further, proposed future infrared limb-sounders will allow to characterize the horizontal and vertical structure of PSCs with considerably improved spatial resolution and coverage. Thus, they would be of great help to gain even more insight in the processes of PSCs in a changing stratospheric environment.

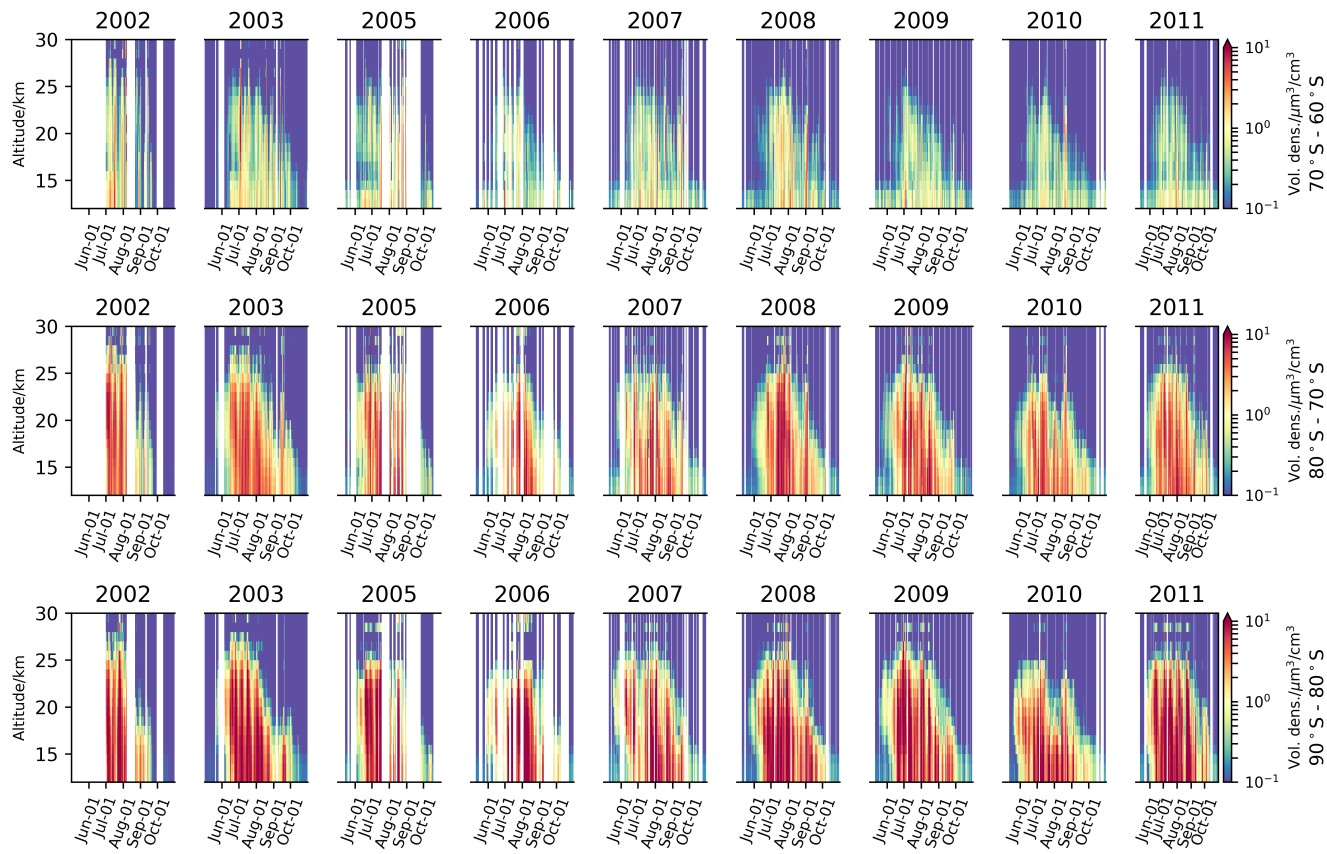

**Figure 8.** Daily zonal mean PSC volume densities for $10°$ latitude bins derived from all MIPAS observations during Antarctic polar winter. The values are averages of the minimum/maximum retrievals as described in the text.

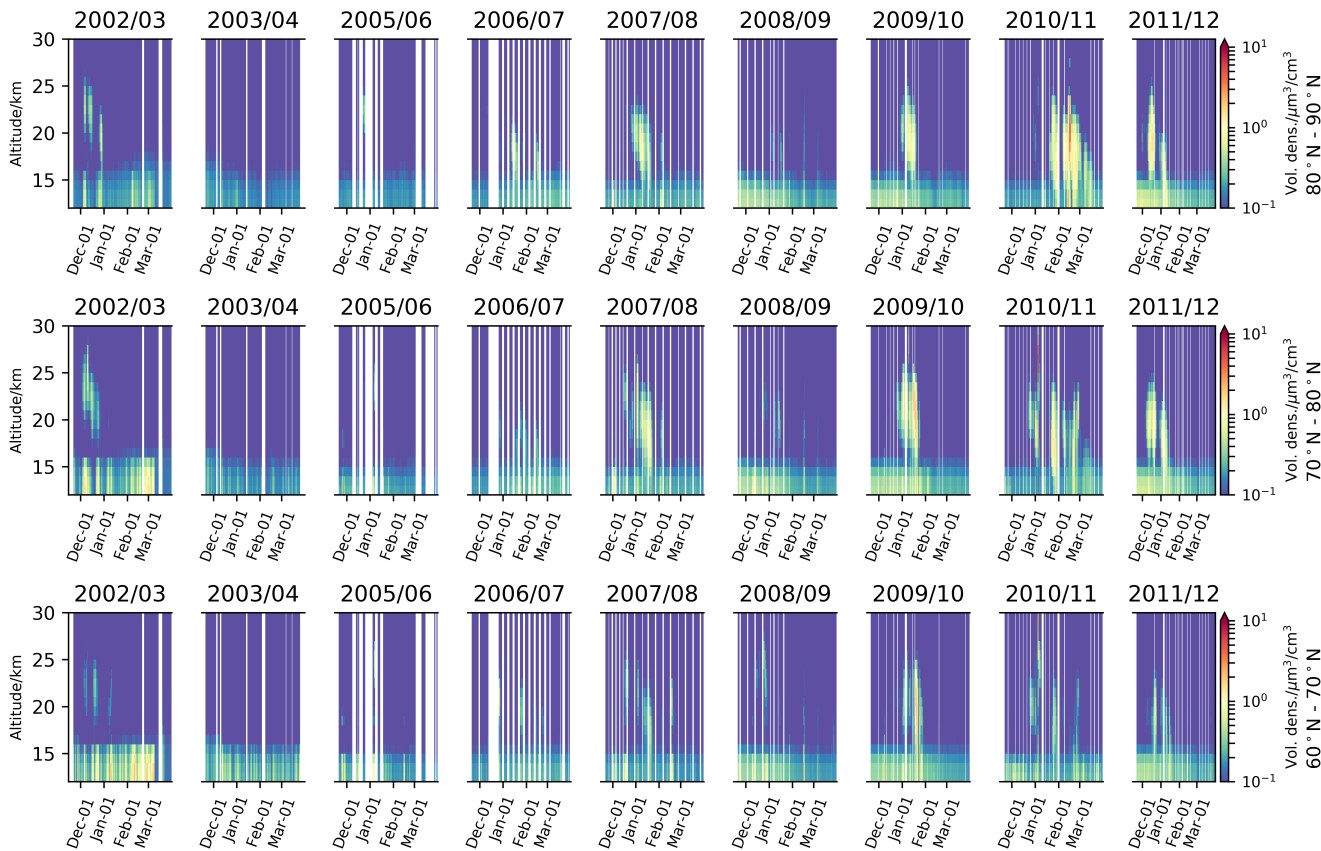

**Figure 9.** Daily zonal mean PSC volume densities for $10°$ latitude bins derived from all MIPAS observations during Arctic polar winter. The values are averages of the minimum/maximum retrievals as described in the text.

*Data availability.* The PSC volume density dataset is available upon request from the author or at http://www.imk-asf.kit.edu/english/308.php.

**Appendix A**

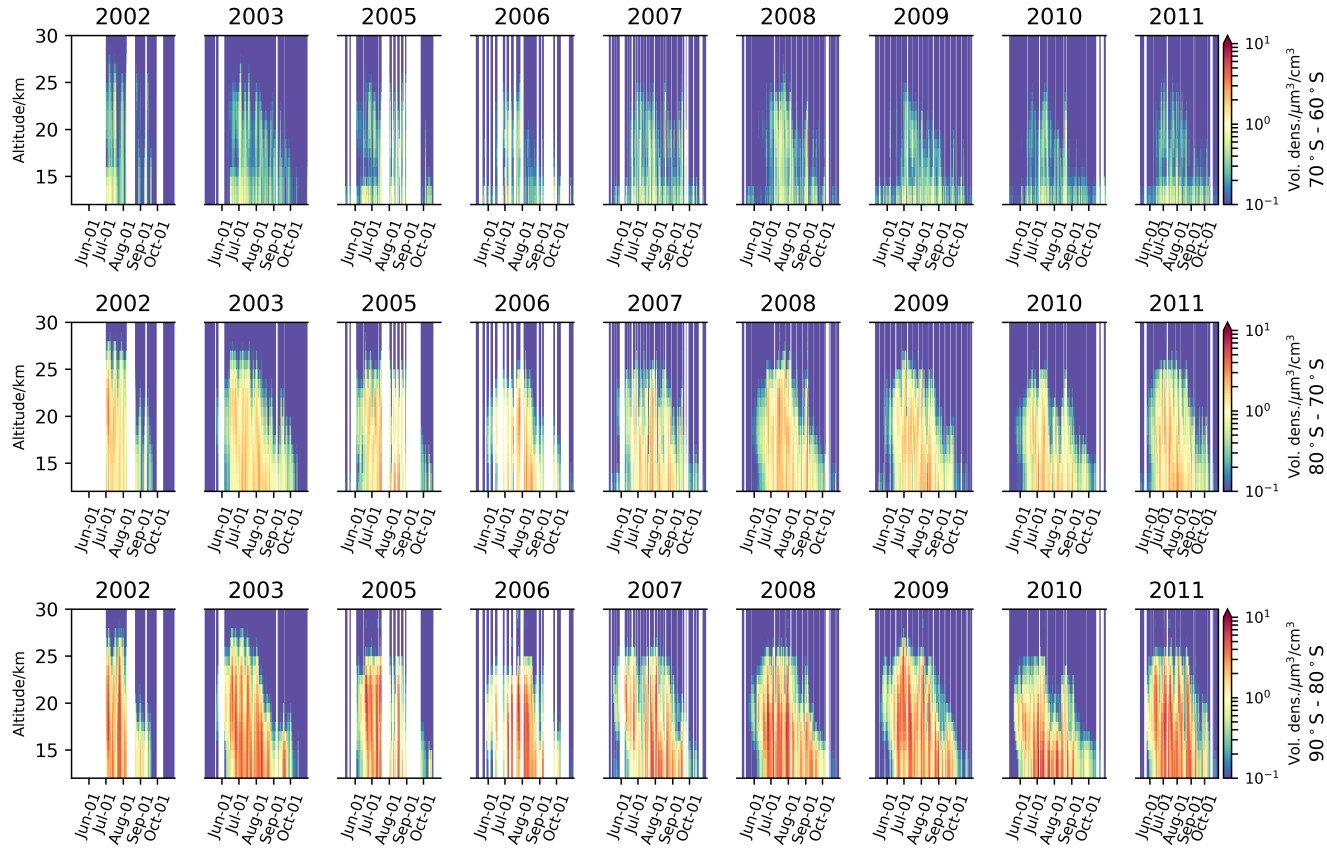

**Figure A1.** Daily zonal minimum PSC volume densities for $10°$ latitude bins derived from all MIPAS observations during Antarctic polar winter. The values are the minimum retrieval limits ($VD_{min}$) as described in the text.

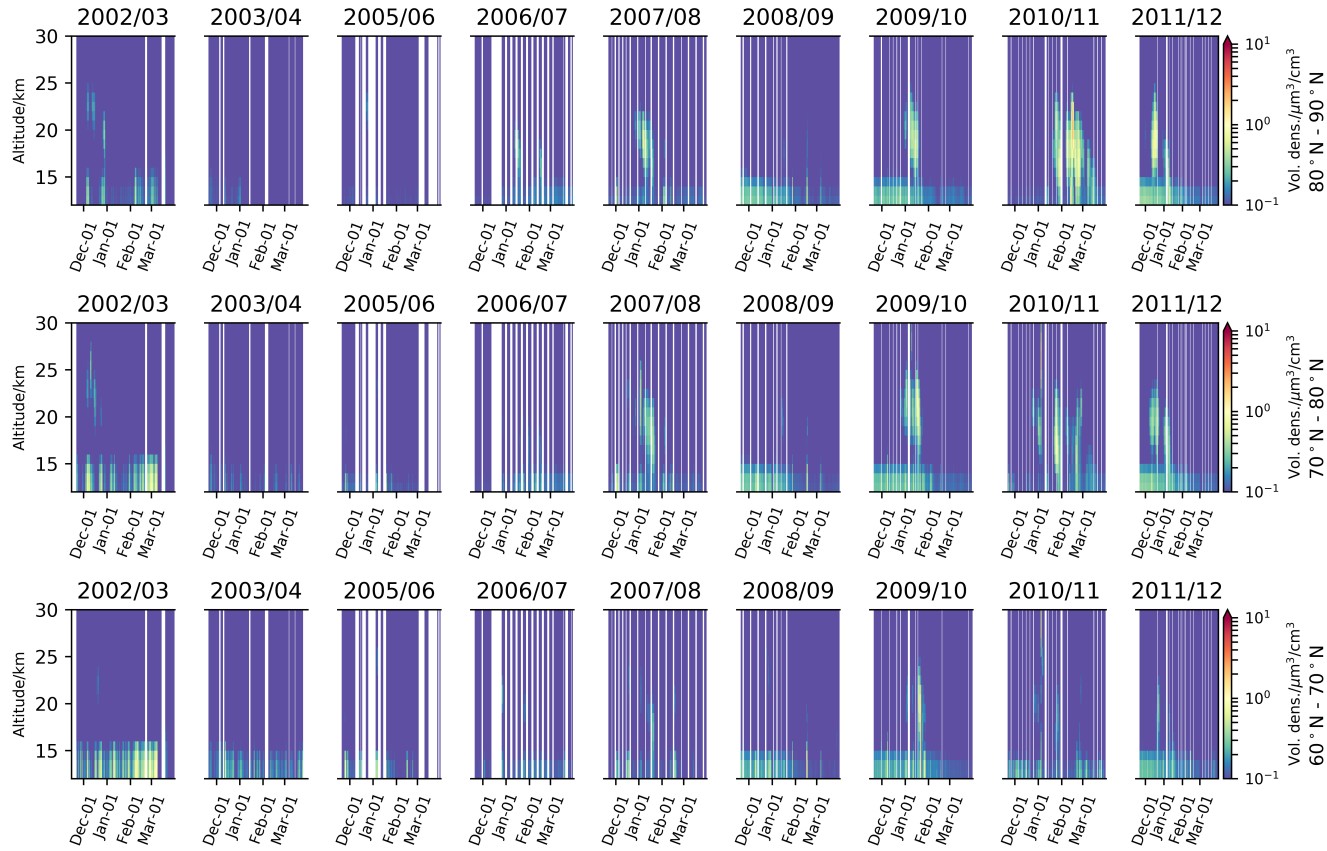

**Figure A2.** Daily zonal minimum PSC volume densities for $10°$ latitude bins derived from all MIPAS observations during Arctic polar winter. The values are the minimum retrieval limits ($VD_{min}$) as described in the text.

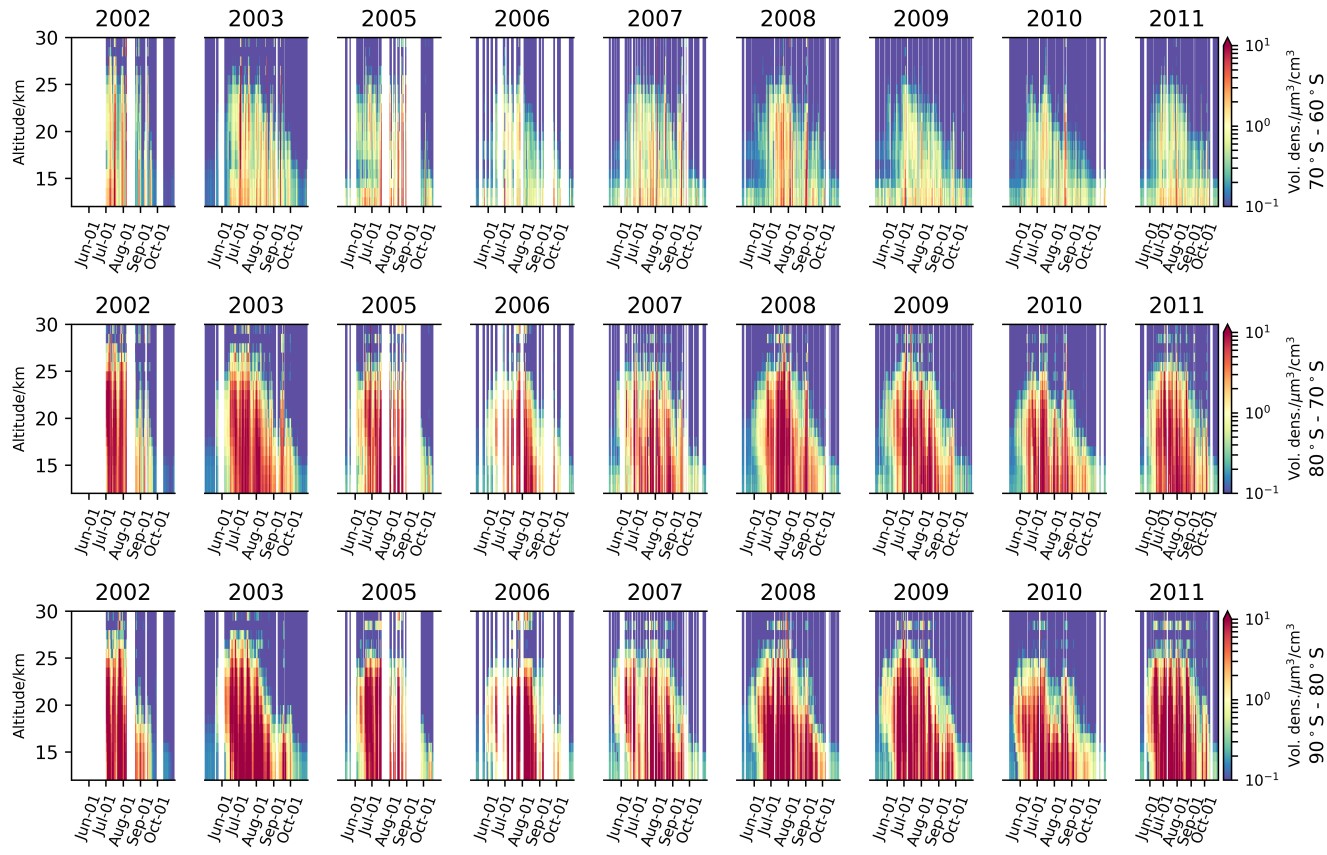

**Figure A3.** Daily zonal maximum PSC volume densities for $10°$ latitude bins derived from all MIPAS observations during Antarctic polar winter. The values are the maximum retrieval limits (VD$_{max}$) as described in the text.

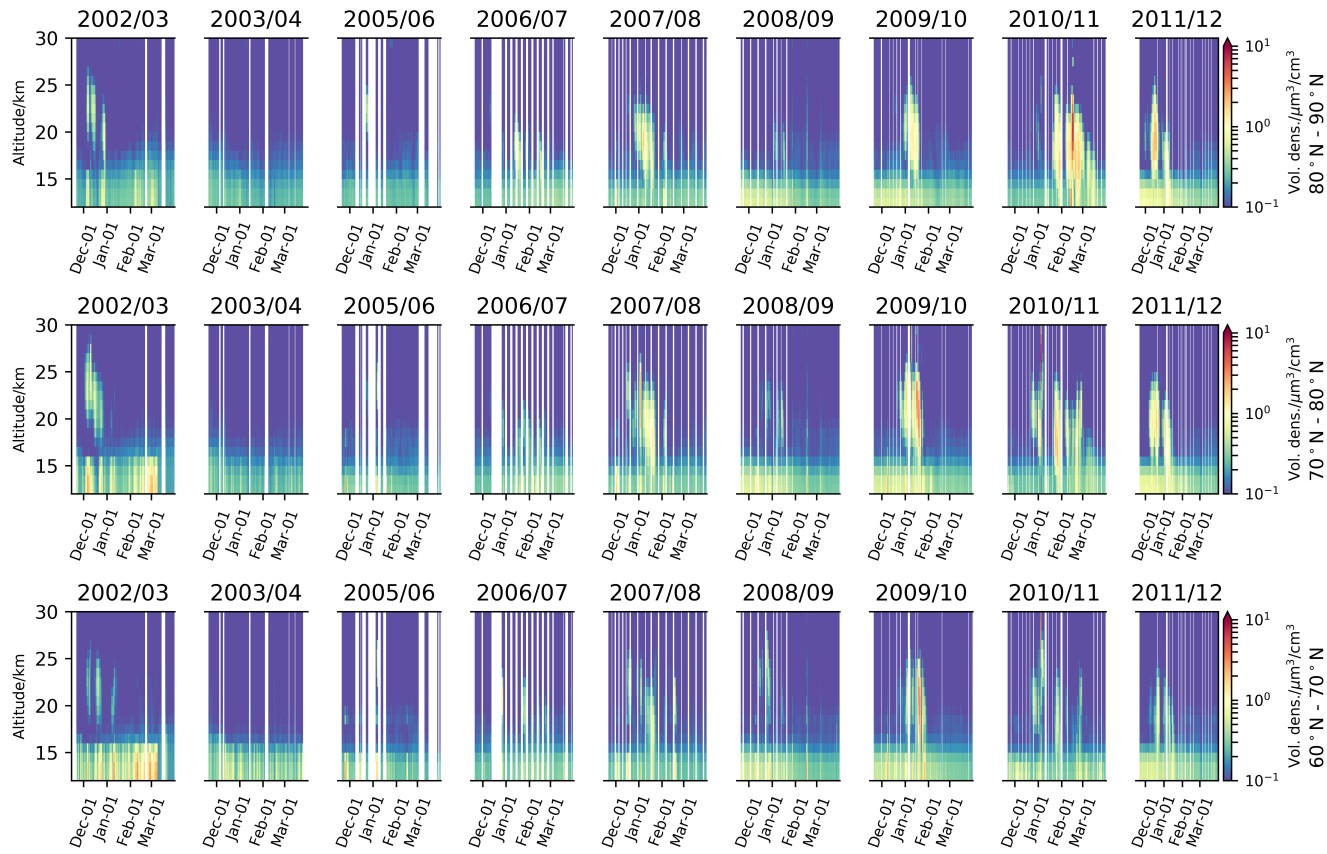

**Figure A4.** Daily zonal maximum PSC volume densities for $10°$ latitude bins derived from all MIPAS observations during Arctic polar winter. The values are the maximum retrieval limits ($VD_{max}$) as described in the text.

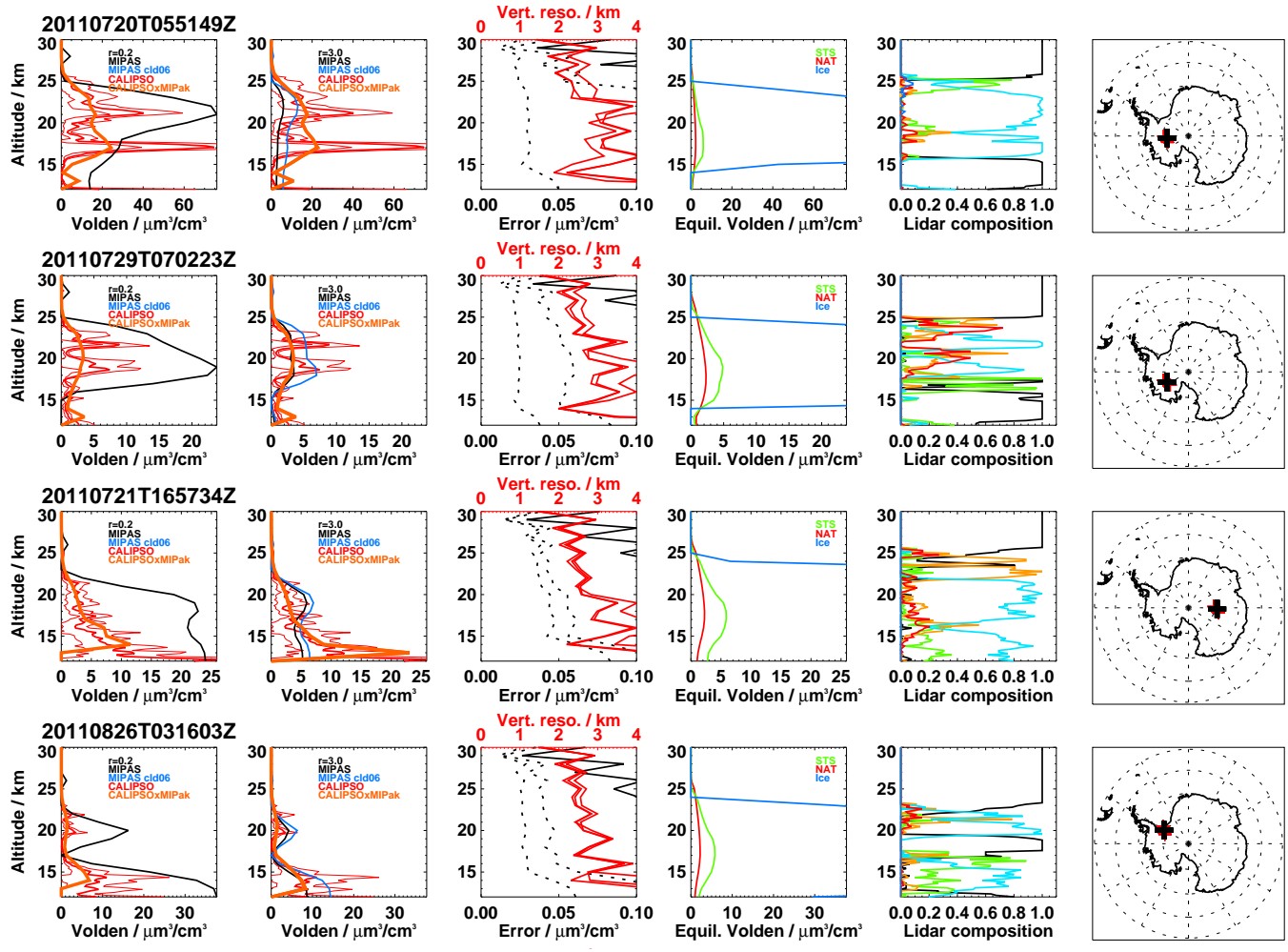

**Figure A5.** Examples of volume density profiles derived from CALIOP and MIPAS for observations within 200 km and 2 h during the Antarctic winter in 2011. Retrieval artifacts at 28 and 26 km altitude are visible in the MIPAS profiles (black lines) in the first column. See caption of Fig. 4 for the content of the columns.

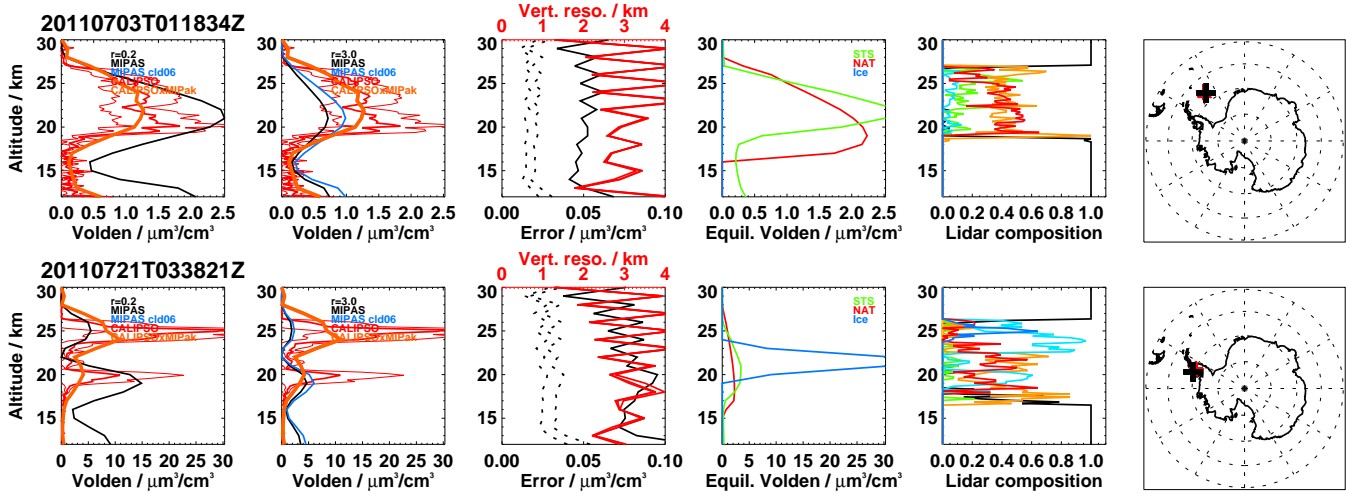

**Figure A6.** Same as Fig. A5 without retrieval instabilities but with enhanced volume densities above 25 km altitude in case of both instruments.

*Competing interests.* The authors declare that they have no conflict of interest.

*Acknowledgements.* The European Space Agency (ESA) is acknowledged for provision of MIPAS level-1b calibrated spectra and for support within the study "Characterisation of particulates in the upper troposphere/lowerstratosphere (ESA Contract No:400011677/16/NL/LvH)". Meteorological data have been provided by the European Centre for Medium-Range Weather Forecasts (ECMWF). This work has fur-
5    ther been supported by the International Space Science Institute (ISSI) and Stratosphere-troposphere Processes And their Role in Climate (SPARC).

We further acknowledge the Deutsche Forschungsgemeinschaft and Open Access Publishing Fund of the Karlsruhe Institute of Technology.

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
