# Peer review of "The MIPAS/Envisat climatology (2002–2012) of polar stratospheric cloud (PSC) volume density profiles"

_Atmospheric Measurement Techniques, 2018_

## Referee Comment (RC1) · Anonymous Referee #2 · 7 Aug 2018

Review of "The MIPAS/Envisat climatology (2002-2012) of polar stratospheric cloud (PSC) volume density profiles" by M. Höpfner et al.

General comments

In this new study, Höpfner et al. introduce a new retrieval scheme for PSC volume densities from the Envisat MIPAS instrument. The scheme is based on a number of simplifying assumptions, e.g., use of NAT refractive indices for all PSC types and neglecting of scattering effects in the radiative transfer calculations. However, the implications of this approximations are thoroughly discussed and seem to be justified. The scheme is used to process retrievals for the entire Envisat mission, and the MIPAS PSC volume

density climatology for the years 2002-2012 is presented.

Overall, this is an interesting study and the paper fits in the scope of AMT. The manuscript is mostly well written and concise. I would recommend it for publication in AMT subject to fixing of some minor comments listed below.

Specific comments

p2, l2-4: Please consider adding a reference for the denitrification process.

p3, l6-7: You are listing the global number of MIPAS vertical scans per day, but how many profiles are measured in the polar regions (which are relevant for PSC observations)?

p4, l17-18: Maybe add reference to Rodgers (2000) for retrieval theory?

p4, l20: I was wondering if you applied a constant Jacobian K or if you considered variations with the state, i.e., $K\_i$?

p5, l21: The link to the ECMWF data is pointing to surface data rather than profiles?

p5, l27-30: Why did you specifically select top altitudes of 6 and 8 km for the low-level clouds?

p7, l9: Which trace gases have been considered in the radiative transfer calculations? Where did their concentrations come from?

p7, l19.23: Why did you not consider using a classification scheme for PSC types (e.g., Spang et al., 2016) instead of selecting the NAT refractive indices for all PSC types?

Spang, R., Hoffmann, L., Höpfner, M., Griessbach, S., Müller, R., Pitts, M. C., Orr, A. M. W., and Riese, M.: A multi-wavelength classification method for polar stratospheric cloud types using infrared limb spectra, Atmos. Meas. Tech., 9, 3619-3639, https://doi.org/10.5194/amt-9-3619-2016, 2016.

p9, Fig. 3: The third plot in the upper row seems to show a rather poor retrieval result,

considering that it refers to a NAT case?

p11, Fig. 4: The caption refers to "PSC volume densities in case of equilibrium". What is this?

p14, Fig. 6: What about the other PSC types, i.e., NAT and ice?

p15, l14-19: Are cirrus clouds really a likely explanation for the enhanced background values in the polar regions? I would not expect to see cirrus clouds up to 15 km of altitude in the polar winter hemisphere.

p15, l24-26: I was wondering how limited the computer resources really are? How many CPU hours were needed to process the entire mission?

p16, l9-10: The new PSC data did not seem to be available at the given web site when I checked the link?

Technical corrections

p8, Fig. 2: x-axis labels have been cropped/clipped.

p13, l15: fix "shows values of than about"

---

## Referee Comment (RC2) · M. Fromm (Referee) · 23 Aug 2018

Reviewer: Mike Fromm

Note: My report includes the pdf of the manuscript annotated with comment bubbles identifying technical issues.

H18 extend the foundational work of Spang et al. (2018) by exploiting the entire MIPAS archive to retrieve PSC volume density (VD). H18 describe the prior MIPAS works on PSC detection and typing, propose to derive VD, a useful quantity for polar processing and chemistry applications, and then show an overview of PSC VD data as a "climatology." H18 is well organized. The section on the retrieval development is clear and thorough. Their retrieval method will be beneficial to the AMT audience.

To the extent that this work's new contribution is in the VD approach, algorithm, and verification with respect to independent PSC data, this manuscript is well targeted to AMT. The work goes on to present some summary PSC VD patterns in the 11-year MIPAS era. Even though the climatology aspect of this work is considerable, and perhaps a better fit for another journal such as ACP, the balance H18 struck between algorithm and applied science permits me to conclude that this is an acceptable candidate for AMT.

That being said, H18 need to motivate the effort they put in to deriving PSC VD. The paper's introduction does a nice job of framing the state of MIPAS PSC developments but does not offer a science reason for the creation of a PSC VD data set. If H18 revise the Introduction to make a compelling case for the value of PSC VD—over and above PSC occurrence and composition (already completed by Spang et al. (2018)—that would provide important motivation to justify the analysis H18 present. Besides that, there are a number of minor and technical issues the authors will need to address before this work can be considered ready for publication in AMT.

Below I list these concerns. In addition, the manuscript has been annotated with comment bubbles identifying specific, technical, and/or grammatical items needing attention. It is provided as part of this report.

Introduction. H18 briefly mention the apparent weakness of prior approaches with limb sounding of IR radiance "...without consideration of the fact that each raypath of the observation intersects multiple altitude levels, leading to an intertwined retrieval problem..." but do not explicitly state how their approach overcomes this weakness. Perhaps this is well articulated in latter portions of the paper, but I was not able to find it. The Introduction needs a statement as to how this is dealt with for the benefit of the science quotient of the new PSC VD data set.

Introduction: Presumably there is added value to the science community to have PSC data expressed in terms of VD. But the reader is not given the argument for this or a literature background on this topic. It would be essential for h18 to make that argument in order to motivate this work.

Figure 1. Three areas of clarification are needed. 1. The plots have up to 3 different lines, solid black, presumably VD; solid orange, presumably median radius; and black dotted line. The caption describes the dotted line as a "first mode" (presumably in size units, which are scaled in orange). I don't see any solid lines of any color that indicate "the second mode." Either more lines are being described than shown, or the descriptions themselves need to be revised. 2. How is one to interpret the ice PSC plots where the VD=0 and median radius is »0. Doesn't VD=0 indicate no PSC? Or does it just indicate no ice? Some explanation would help. 3. The 19920127 STS plot shows a very large median radius. Why is that classified as STS?

P15, L5-6. H18 make the point that the MIPAS data shown in Figure 7 are unique because measures much closer to the pole than any other satellite PSC data set. This point is well taken, but the onset date they show is not notably different than Antarctic onset dates recorded by instruments farther from the pole (e.g. CALIPSO, SAM II, POAM II, III). The advantage of MIPAS is that it provides this uniquely near-pole coverage throughout the season, in both hemispheres (as Spang et al. 2018 point out). Perhaps H18 might consider enhancing/refining the discussion here?

Abstract: Related to the point made above, H18 make a statement in the abstract about "this climatology captures this onset. . ." However, isn't it the more general MIPAS PSC data set and climatology (previously reported) that gets the credit for this rather than the specific VD climatology? If in fact the newly developed PSC VD data set shines a unique light on this, the paper needs to make clear how the VD data set expands the constraints to higher latitudes and different times than the MIPAS PSC-detection affords.

P15, L11 (and Figure 8). "single enhanced values are visible" The panels of Figure 8 are very small and there are white bars competing with the VD color scale, making the features called out difficult to discern. Perhaps H18 could elaborate on what they mean by the quote. Perhaps also they could provide a single, expanded panel with the feature of note pointed out. Finally, it is not evident to me that PSC features above 27 km must be artificial. PSCs have been observed higher than 27 km regularly (CALIPSO curtains show this to be true). Hence I'd ask H18 to consider enhancing the visualization of this feature, describing it more clearly, and discussing whether it might also be evidence of real PSCs in addition to the "side-lobe" feature they identify.

P13, L19-20. "Here, both MIPAS retrievals and the CALIOP dataset often indicate much smaller values" Much smaller than what? Please clarify.

Please also note the supplement to this comment:
https://www.atmos-meas-tech-discuss.net/amt-2018-163/amt-2018-163-RC2-supplement.pdf

**Supplement:**

**The MIPAS/Envisat climatology (2002–2012) of polar stratospheric cloud (PSC) volume density profiles**

Michael Höpfner[1], Terry Deshler[2], Michael Pitts[3], Lamont Poole[4], Reinhold Spang[5], Gabriele Stiller[1], and Thomas von Clarmann[1]

[1]Institute of Meteorology and Climate Research, Karlsruhe Institute of Technology, Karlsruhe, Germany
[2]Department of Atmospheric Science, University of Wyoming, Laramie, Wyoming, USA
[3]NASA Langley Research Center, Hampton, Virginia, USA
[4]Science Systems and Applications, Incorporated, Hampton, Virginia, USA
[5]Institut für Energie und Klimaforschung, Stratosphäre, IEK-7, Forschungszentrum Jülich, Jülich, Germany

**Correspondence:** M. Höpfner (michael.hoepfner@kit.edu)

**Abstract.** A global data set of vertical profiles of polar stratospheric cloud (PSC) volume densities has been derived from Michelson Interferometer for Passive Atmospheric Sounding (MIPAS) space-borne infrared limb measurements between 2002 and 2012. To develop a well characterized and efficient retrieval scheme, systematic tests based on limb-radiance simulations for PSCs from in-situ balloon observations have been performed . The finally selected wavenumber range was around

5   $830 \, \mathrm{cm}^{-1}$. Optical constants of nitric acid trihydrate (NAT) have been used to derive maximum and minimum profiles of volume density which are compatible with MIPAS observations under the assumption of small, non-scattering and larger, scattering PSC particles. These max/min profiles deviate from their mean value at each altitude by about 40-45%, which is attributed as the maximum systematic error of the retrieval. Further, the retrieved volume density profiles are characterized by a random error due to instrumental noise of 0.02–0.05 $\mu\mathrm{m}^3\mathrm{cm}^{-3}$, a detection limit of about 0.1–0.2 $\mu\mathrm{m}^3\mathrm{cm}^{-3}$ and a vertical

[revised manuscript text omitted]

---

## Short Comment (SC1) · 12 Sep 2018

Höpfner et al. present a global data set of vertical profiles of volume densities of PSCs. They have derived their data from MIPAS measurements. They use beta-NAT optical constants in the wavenumber range around 830 cm-1 in order to interpret the profiles of volume density. Strong variability of PSC parameters in different Artic stratosphere winters has been observed.

Unfortunately, the authors ignore the fact that more than one NAT phase (i.e. alpha and beta NAT) is known from literature [1] and that both phases in combination with ice can occur in the lower polar stratosphere [2]. For both phases, optical constants

have not only been calculated, but have also been measured in the laboratory with high precision [3, 4]. The morphology of the crystalline particles can have an important impact on the spectra as well [5, 6].

Beside NAT also NAD is a possible phase in PSCs [7ab]. Also NAD exhibits two crystalline modifications (alpha and beta NAD) [8]. Cold chamber experiments show that the metastable low temperature phase is more likely [9]. The authors should include these latest spectroscopic and mechanistic results from the literature into their discussion. Eventually, this will help to understand the reported large variabilities.

In addition to the points raised above it behooves the authors to compare the complex index of refraction with absorption cross section data recently published in the literature [4]. The authors should use information from laboratory experiments to calibrate, or at least validate the field observations using independent verification. Retrievals do not mean anything unless the optical data are validated using information external to the retrieval cycle. Table 6 of reference 4 displays quantitative data on optical constants of alpha- and beta-NAT as well as NAD (exact phase unknown). The associated spectra also show that all three nitric acid hydrates absorb around 830 cm-1 such that basing the assessment of the occurrence frequency solely on a single wavelength region is a losing proposition. It behooves the authors to make better use of published results from laboratory experiments in order to maximize the scientific insight (phase, frequency of occurrence, interconversion dynamics and the like) to the benefit of the readers who will appreciate a break-out from the beaten path from time to time.

References

[1] H. Tizek, E. Knözinger, H. Grothe: "Formation and Phase Distribution of Nitric Acid Hydrates in the Mole Fraction Range xHNO3 < 0.25: a combined XRD and IR study"; Physical Chemistry Chemical Physics, 6 (2004), 972 - 979.

[2] F. Weiss, F. Kubel, O. Galvez, M. Hölzel, S. F. Parker, P. Baloh, R. Iannarelli, M.J. Rossi, H. Grothe: "Metastable Nitric Acid Trihydrate in Ice Clouds"; Angewandte

Chemie - International Edition, 55 (2016), 10; 3276 - 3280.

[3] I.K. Ortega, B. Maté, M.A. Moreno, V.J. Herrero, and R. Escribano: "Infrared spectra of nitric acid trihydrate (beta-NAT): A comparison of available optical constants and implication for the detection of polar stratospheric clouds (PSC's), Geophys. Res. Lett., 33, (2006), L19816, doi:10.1029/2006GL026988.

[4] R. Iannarelli, M.J. Rossi, "The mid-IR Absorption Cross Sections of $\alpha$- and $\beta$-NAT (HNO3-3H2O) in the range 170 to 185K and of metastable NAD (HNO3-2H2O) in the range 172 to 182K" Journal of Geophysical Research Atmospheres 120 (2015), 11707-11727.

[5] H. Grothe, H. Tizek, I. Ortega: "Metastable Nitric Acid Hydrates - Possible Constituents of Polar Stratospheric Clouds?"; Faraday Discussions, 137 (2008), 223 - 234.

[6] H. Grothe, H. Tizek, D Waller, D Stokes: "The Crystallization Kinetics and Morphology of Nitric Acid Trihydrate"; Physical Chemistry Chemical Physics, 8 (2006), 2232 - 2239.

[7a] O. Stetzer, O. Möhler, R. Wagner, S. Benz, H. Saathoff, H. Bunz, O. Indris: "Homogeneous nucleation rates of nitric acid dihydrate (NAD) at simulated stratospheric conditions–Part I: Experimental results" Atmospheric Chemistry and Physics 6, 10, (2006), 3023-3033.

[7b] O. Möhler, H. Bunz, O. Stetzer: "Homogeneous nucleation rates of nitric acid dihydrate (NAD) at simulated stratospheric conditions–Part II: Modelling" Atmospheric Chemistry and Physics 6, 10, (2006), 3035-3047.

[8] H. Tizek, E. Knözinger, H. Grothe:"X-ray diffraction studies on nitric acid dihydrate"; Physical Chemistry Chemical Physics, 4 (2002), 5128 - 5134.

[9] R. Wagner, O. Möhler, H. Saathoff, O. Stetzer, U. Schurath: "Infrared spectrum of nitric acid dihydrate: Influence of particle shape" The Journal of Physical Chemistry A 109, 11, (2005), 2572-2581.

---

## Author Comment (AC1) · 5 Oct 2018

We thank referee # 2 for the very valuable comments and corrections. Our answers are given below. The original referee comment is repeated in **bold**, changes in the manuscript text are printed in *italic*.

**General comments**

**In this new study, Höpfner et al. introduce a new retrieval scheme for PSC volume densities from the Envisat MIPAS instrument. The scheme is based on a number of simplifying assumptions, e.g., use of NAT refractive indices for all PSC types**

**and neglecting of scattering effects in the radiative transfer calculations. However, the implications of this approximations are thoroughly discussed and seem to be justified. The scheme is used to process retrievals for the entire Envisat mission, and the MIPAS PSC volume density climatology for the years 2002-2012 is presented.**

**Overall, this is an interesting study and the paper fits in the scope of AMT. The manuscript is mostly well written and concise. I would recommend it for publication in AMT subject to fixing of some minor comments listed below.**

**Specific comments**

**p2, l2-4: Please consider adding a reference for the denitrification process.**

We have added references to Fahey et al. (1990) and to the review article of Solomon (1999).

*Fahey, D. W., Kelly, K. K., Kawa, S. R., Tuck, A. F., Loewenstein, M., Chan, K. R., and Heidt, L. E.: Observations of denitrification and dehydration in the winter polar stratospheres, Nature, 344, 321–324, https://doi.org/10.1038/344321a0, 1990.*

*Solomon, S.: Stratospheric ozone depletion: A review of concepts and history, Rev. Geophys., 37, 275–316, https://doi.org/10.1029/1999RG900008, 1999.*

**p3, l6-7: You are listing the global number of MIPAS vertical scans per day, but how many profiles are measured in the polar regions (which are relevant for PSC observations)?**

We agree that these numbers are also relevant and have appended the text accordingly:

*In regions poleward of 60° latitude about 170 and 240 profiles per day have been obtained during each period, respectively.*

**p4, l17-18: Maybe add reference to Rodgers (2000) for retrieval theory?**

We will add this reference.

*Rodgers, C. D.: Inverse Methods for Atmospheric Sounding: Theory and Practice, Vol. 2 of Series on Atmospheric, Oceanic and Planetary Physics, World Scientific, 2000.*

**p4, l20: I was wondering if you applied a constant Jacobian K or if you considered variations with the state, i.e., $K_i$?**

The Jacobians depend on the iteration. Thus, the correct notation should have been $K_i$. This will be revised.

**p5, l21: The link to the ECMWF data is pointing to surface data rather than profiles?**

The link has been corrected to http://apps.ecmwf.int/datasets/data/interim-full-daily/levtype=pl/.

**p5, l27-30: Why did you specifically select top altitudes of 6 and 8 km for the low-level clouds?**

This choice was driven by the wish to introduce scenes in-between the two extreme cases of having scattering with no cloud below ("scat_nocld" in Fig. 2), which results in the highest limb radiances and no scattering ("noscat"), leading to the lowest limb radiances. To set 8 km as the highest tropospheric cloud was motivated in order keep an altitude separation between tropospheric clouds and PSCs from the in-situ dataset (reaching down to 10 km altitude). The 6 km cloud case was chosen as an intermediate between "cld8" and "nocld" since it often lead to retrieval errors half between those two adjacent cases (see Fig. 2).

**p7, l9: Which trace gases have been considered in the radiative transfer calculations? Where did their concentrations come from?**

The information about the trace gases simulated, has been added in the new version of the manuscript:

*The following trace gases have been considered within the radiative transfer simulations: $H_2O$, $CO_2$, $O_3$, $N_2O$, $CH_4$, $HNO_3$, $C_2H_6$, CFC-11, CFC-22.*

The information about the source of the concentration altitude profiles is already provided on p. 5, l. 26 of the AMTD version: "Trace gas profiles are obtained from polar winter standard atmospheres (Remedios et al., 2007)."

**p7, l19.23: Why did you not consider using a classification scheme for PSC types (e.g., Spang et al., 2016) instead of selecting the NAT refractive indices for all PSC types? Spang, R., Hoffmann, L., Höpfner, M., Griessbach, S., Müller, R., Pitts, M. C., Orr, A. M. W., and Riese, M.: A multi-wavelength classification method for polar stratospheric cloud types using infrared limb spectra, Atmos. Meas. Tech., 9, 3619-3639, https://doi.org/10.5194/amt-9-3619-2016, 2016.**

We have decided not to use a-priori data on composition in the retrieval process but rather to develop a retrieval set-up which is as far as possible robust with respect to composition (and particle size). This has been motivated by the observations that very often PSCs do not consist of a single type (see. e.g. Pitts et al., 2018, Fig. 10, https://doi.org/10.5194/acp-18-10881-2018). Especially given the large field-of-view volume a limb-sounder like MIPAS is covering in one observation (3-4 km in the vertical, 30 km horizontal across-track and several 100 km along track), frequently PSC particles of different composition will contribute to the spectral radiances. Further, in the MIPAS composition classification like in Spang et al. (2018) (https://doi.org/10.5194/acp-18-5089-2018) there are still ambiguities between e.g. STS and large NAT particles which could increase the uncertainty of a retrieval depending on exact composition information further.

**p9, Fig. 3: The third plot in the upper row seems to show a rather poor retrieval result, considering that it refers to a NAT case?**

In the text (p. 7, l. 28) with regard to this Figure it is explained: "In case of NAT as the predominant composition, the volume densities are generally overestimated, since

scattering is neglected in the retrieval. It can well be observed that the less scattering contributes from the troposphere, which is the case for a cold tropospheric cloud at 8 km, the better the result fits the reference". This means that in spite of the fact that NAT refractive indices are used, there is still the uncertainty of the particles size leading, through the unknown amount of scattered radiation, to the uncertainties in the retrieved profiles. As this has been shown to be the largest error contribution, we have decided to provide minimum/maximum profiles as the result of our MIPAS retrievals.

**p11, Fig. 4: The caption refers to "PSC volume densities in case of equilibrium". What is this?**

Right, an explanation is missing here. Therefore we have changed the Figure caption to:

*The forth column contains the PSC volume densities in case of thermodynamic equilibrium (see text).*

In the main text (p. 13, l. 17), we have added an explanation:

*Retrieved profiles of particle volume densities can be compared to the volume, solid or liquid PSC phases can reach under thermodynamic equilibrium conditions (Hanson and Mauersberger, 1988; Carslaw et al., 1994). We have calculated these profiles using temperatures from ECMWF, standard polar winter concentration profiles of $HNO_3$ and $H_2O$ (Remedios et al., 2007) and 0.3 ppbv of $H_2SO_4$.*

*Hanson, D. and Mauersberger, K.: Laboratory studies of the nitric acid trihydrate: Implications for the south polar stratosphere, Geophys. Res. Lett., 15, 855–858, https://doi.org/10.1029/88GL00209, 1988.*

*Carslaw, K. S., Luo, B. P., Clegg, S. L., Peter, T., Primblecombe, P., and Crutzen, P. J.: Stratospheric aerosol growth and HNO3 gas phase depletion from coupled HNO3 and water uptake by liquid particles, Geophys. Res. Lett., 21, 2479–2482, https://doi.org/10.1029/94GL02799, 1994.*

*Remedios, J. J., Leigh, R. J., Waterfall, A. M., Moore, D. P., Sembhi, H., Parkes, I., Greenhough, J., Chipperfield, M. P., and Hauglustaine, D.: MIPAS reference atmospheres and comparisons to V4.61/V4.62 MIPAS level 2 geophysical data sets, Atmos. Chem. Phys. Discuss., 7, 9973–10 017, https://doi.org/10.5194/acpd-7-9973-2007, https://www.atmos-chem-phys-discuss.net/7/9973/2007/, 2007.*

**p14, Fig. 6: What about the other PSC types, i.e., NAT and ice?**

As we have mentioned in the manuscript (p. 10, l. 10): "... (Pitts et al., 2018). Their estimated uncertainties of volume density derived in case of STS PSCs are in the range of 0.05–1.0 $\mu m^3/cm^3$. For NAT mixtures and ice PSCs, the CALIOP volume density values are mostly lower limits and can be underestimated by factors of 10 and up to 30 for NAT and ice PSCs, respectively". Thus, unlike for STS, we do not think that systematic comparisons between CALIPSO and MIPAS retrievals in case of NAT and ICE PSCs are of any help to assess the accuracy of our MIPAS dataset. Nonetheless, in Figures 4 and 5 we have plotted single comparisons also for NAT and ICE which already show the difficulties in comparing those observations.

**p15, l14-19: Are cirrus clouds really a likely explanation for the enhanced background values in the polar regions? I would not expect to see cirrus clouds up to 15 km of altitude in the polar winter hemisphere.**

Due to the limited vertical resolution of MIPAS we cannot exclude that cirrus clouds (which have a much larger optical thickness in limb direction than PSCs) at 12-13 km altitude may influence the retrievals up to 15 km. To make this point clearer, we have slightly modified the text:

*... by the influence of tropospheric cirrus, which e.g. at 12-13 km reach the lower edge of the vertical field-of-view of MIPAS pointing at 14-15 km tangent height.*

**p15, l24-26: I was wondering how limited the computer resources really are? How many CPU hours were needed to process the entire mission?**

Considering this comment, we have come to the conclusion that the related text passage "... to develop a retrieval approach applicable within limited computer resources" is not entirely convincing given the large variability of computer capacities at different institutions. We have therefore decided to skip this part.

**p16, l9-10: The new PSC data did not seem to be available at the given web site when I checked the link?**

The data is available now.

**Technical corrections**

**p8, Fig. 2: x-axis labels have been cropped/clipped.**

Corrected.

**p13, l15: fix "shows values of than about"**

Corrected.

---

## Author Comment (AC2) · 5 Oct 2018

Many thanks to Michael Fromm for his thorough review helping to improve the manuscript substantially.

Our answers are given below. The original referee comment is repeated in **bold**, changes in the manuscript text are printed in *italic*.

**Note: My report includes the pdf of the manuscript annotated with comment bubbles identifying technical issues.**

All comments and suggestions identified in the pdf have been addressed/incorporated

in the revised manuscript.

**H18 extend the foundational work of Spang et al. (2018) by exploiting the entire MIPAS archive to retrieve PSC volume density (VD). H18 describe the prior MIPAS works on PSC detection and typing, propose to derive VD, a useful quantity for polar processing and chemistry applications, and then show an overview of PSC VD data as a "climatology." H18 is well organized. The section on the retrieval development is clear and thorough. Their retrieval method will be beneficial to the AMT audience.**

**To the extent that this work's new contribution is in the VD approach, algorithm, and verification with respect to independent PSC data, this manuscript is well targeted to AMT. The work goes on to present some summary PSC VD patterns in the 11-year MIPAS era. Even though the climatology aspect of this work is considerable, and perhaps a better fit for another journal such as ACP, the balance H18 struck between algorithm and applied science permits me to conclude that this is an acceptable candidate for AMT.**

**That being said, H18 need to motivate the effort they put in to deriving PSC VD. The paper's introduction does a nice job of framing the state of MIPAS PSC developments but does not offer a science reason for the creation of a PSC VD data set. If H18 revise the Introduction to make a compelling case for the value of PSC VD over and above PSC occurrence and composition (already completed by Spang et al. (2018). That would provide important motivation to justify the analysis H18 present. Besides that, there are a number of minor and technical issues the authors will need to address before this work can be considered ready for publication in AMT.**

We agree that the introduction does not contain much information about the motivation for a specific VD retrieval (in fact, the information is spread in the later sections). Below, under the specific comments, we explain how the new manuscript will be revised

accordingly.

**Below I list these concerns. In addition, the manuscript has been annotated with comment bubbles identifying specific, technical, and/or grammatical items needing attention. It is provided as part of this report.**

**Introduction. H18 briefly mention the apparent weakness of prior approaches with limb sounding of IR radiance "...without consideration of the fact that each raypath of the observation intersects multiple altitude levels, leading to an intertwined retrieval problem..." but do not explicitly state how their approach overcomes this weakness. Perhaps this is well articulated in latter portions of the paper, but I was not able to find it. The Introduction needs a statement as to how this is dealt with for the benefit of the science quotient of the new PSC VD data set.**

Yes, the adopted solution to this problem did not become clear here. What we mean is that, due to the limb geometry, all the previous methods related to occurrence and composition used to investigate MIPAS limb data up to now are mostly valid for the highest PSC-layers and become more and more uncertain at lower altitudes. That is, because in those methods each limb-view is analyzed separately. In case of the presented VD retrieval, we use the information from all limb-views of a whole limb-scan together (like in trace-gas retrievals). Thus, also the lower layers of a PSC should be described better by the VD profiles.

The text will be appended accordingly:

*In the present work we tackle this problem by adopting a complete altitude-resolved inversion of all limb views simultaneously. This means that, like in the case of standard trace gas retrievals, a global fit approach is used to derive altitude profiles of PSC volume densities (e.g. Höpfner et al., 2006b).*

**Introduction: Presumably there is added value to the science community to have**

**PSC data expressed in terms of VD. But the reader is not given the argument for this or a literature background on this topic. It would be essential for h18 to make that argument in order to motivate this work.**

For a better motivation, we have extended the introduction by:

*Beyond the PSC existence and composition, which is already available from MIPAS (Spang et al., 2018), volume density is an independent quantitative parameter which can be used for validation and analysis of atmospheric model results. For example, by comparison with MIPAS data on volume density, Khosrawi et al. (2018) could show that their global model simulates PSC existence well but underestimates strongly the PSC mass which might influence vertical redistribution of $HNO_3$.*

**Figure 1. Three areas of clarification are needed. 1. The plots have up to 3 different lines, solid black, presumably VD; solid orange, presumably median radius; and black dotted line. The caption describes the dotted line as a "first mode" (presumably in size units, which are scaled in orange). I don't see any solid lines of any color that indicate "the second mode." Either more lines are being described than shown, or the descriptions themselves need to be revised. 2. How is one to interpret the ice PSC plots where the VD=0 and median radius is >0. Doesn't VD=0 indicate no PSC? Or does it just indicate no ice? Some explanation would help. 3. The 19920127 STS plotshows a very large median radius. Why is that classified as STS?**

1) We agree that the explanation in the Figure caption should be improved, as well as the visibility of the lines in the plots – they all contain four lines. To improve the latter, we have strongly increase the line thickness. The caption now reads:

*Example profiles from the in-situ balloon database on PSCs used as input for the radiative transfer model. The database contains parameters of bi-modal log-normal distributions derived from the particle counter measurements. Here the median radius (top orange axis) and the total particle volume density (VD, bottom black axis) of each*

*mode are shown. Dotted lines indicate the first mode with smaller particles and solid lines the second mode. The title indicates the date of the balloon observation and the predominant composition of the PSCs (MIX is a mixture of similar volume densities of STS and NAT).*

2) In case of the ICE-PSCs at higher altitudes, the volume densities of the second mode are in the order of 0.1 $\mu$m$^3$/cm$^3$ while the radius is about 1-2 $\mu$m. Thus, there have been particles of that size, but very few, such that the resulting volume is small.

3) It shows a large median radius of the 2nd mode (up to 4.5 $\mu$m), however with a low volume density (about 0.3 $\mu$m$^3$/cm$^3$) compared to the first mode (up to 4 $\mu$m$^3$/cm$^3$). So there might be a few large particles (e.g. NAT), but the volume is dominated by the small particles.

**P15, L5-6. H18 make the point that the MIPAS data shown in Figure 7 are unique because measures much closer to the pole than any other satellite PSC data set. This point is well taken, but the onset date they show is not notably different than Antarctic onset dates recorded by instruments farther from the pole (e.g. CALIPSO, SAM II, POAM II, III). The advantage of MIPAS is that it provides this uniquely nearpole coverage throughout the season, in both hemispheres (as Spang et al. 2018 point out). Perhaps H18 might consider enhancing/refining the discussion here?**

In this paragraph, we intended to present an interesting example of the new MIPAS PSC dataset. In the southern hemisphere, we believe that the first period of PSC evolution is extremely relevant due to the nucleation of NAT, which is still an unresolved problem (e.g. Hoyle et al., 2013), and which leads to the subsequent denitrification of the stratosphere. To make this clearer we have extended the text as:

"As an example for the coverage of single profiles, Fig. 7 shows the retrieved PSC volume densities at 20 km altitude in mid-May 2010. One can clearly observe the onset of PSCs evolution right in the center close to the South Pole. *The appearance and*

*mass of PSCs over Antarctica in May can deliver valuable information on the nucleation
process of NAT particles, which are relevant for denitrification (e.g. Lambert et al.,
2016). For example, in Lambert et al. (2016, Tab. 2) the reported on-set date as
derived from CALIOP is 22-May while according to MIPAS observations (Fig. 7) first
PSCs appear 5–6 days earlier.* During several of the years observed by MIPAS, the
first PSCs are detected in this region during very similar times. These observations
are unique since no other instrument has observed PSCs during their formation so far
south *(Spang et al., 2018)*."

**Abstract: Related to the point made above, H18 make a statement in the abstract
about "this climatology captures this onset . . ." However, isn't it the more general
MIPAS PSC data set and climatology (previously reported) that gets the credit for
this rather than the specific VD climatology? If in fact the newly developed PSC
VD data set shines a unique light on this, the paper needs to make clear how the
VD data set expands the constraints to higher latitudes and different times than
the MIPAS PSC-detection affords.**

We agree that the detection of the PSC on-set alone has already been captured by the
previous MIPAS datasets. Thus, we have changed the related sentence in the abstract
to:

*Among other interesting features, this climatology helps to study quantitatively the on-
set of PSC formation very near to the South Pole and the large variability of the PSC
volume densities between different Arctic stratospheric winters.*

**P15, L11 (and Figure 8). "single enhanced values are visible" The panels of Fig-
ure 8 are very small and there are white bars competing with the VD color scale,
making the features called out difficult to discern. Perhaps H18 could elaborate
on what they mean by the quote. Perhaps also they could provide a single, ex-
panded panel with the feature of note pointed out. Finally, it is not evident to
me that PSC features above 27 km must be artificial. PSCs have been observed**

**higher than 27 km regularly (CALIPSO curtains show this to be true). Hence I'd ask H18 to consider enhancing the visualization of this feature, describing it more clearly, and discussing whether it might also be evidence of real PSCs in addition to the "side-lobe" feature they identify.**

We have expanded the related explanation in the text accordingly. In addition, we have added two figures (see Figures A5 and A6 in the supplement to this comment) in the appendix with MIPAS-CALIOP co-incident comparisons showing the effect of retrieval instabilities compared to high-cloud observations.

*In the plots of the southern hemisphere (bottom two rows of Fig. 8) at altitudes of 28 km and 26 km, bands of enhanced values are visible during mid-winter. These often appear as side-lobes in the retrieved profile when optically thick ice clouds are present, as can be observed in co-incident observations of CALIOP and MIPAS (Fig. A5). In comparison, high-altitude PSCs are mostly not confined to a single retrieval level, visible in Fig. A6. The instabilities could be suppressed by increasing the regularization strength, however, at the expense of a deterioration of the vertical resolution. We have, thus, decided not to change the constraint, but to point at these potential outliers.*

**P13, L19-20. "Here, both MIPAS retrievals and the CALIOP dataset often indicate much smaller values" Much smaller than what? Please clarify.**

We have changed the text to better clarify this point:

*Here, both, MIPAS retrievals and the CALIOP dataset often indicate much smaller values of volume density compared to the calculations under the assumption of thermodynamic equilibrium.*

**Please also note the supplement to this comment: https://www.atmos-meas-tech-discuss.net/amt-2018-163/amt-2018-163-RC2-supplement.pdf**

As mentioned above, all comments in the supplement have been addressed in the new version of the manuscript.

References:

Höpfner, M., Luo, B. P., Massoli, P., Cairo, F., Spang, R., Snels, M., Di Donfrancesco, G., Stiller, G., von Clarmann, T., Fischer, H., and Biermann, U.: Spectroscopic evidence for NAT, STS, and ice in MIPAS infrared limb emission measurements of polar stratospheric clouds, Atmos. Chem. Phys., 6, 1201–1219, https://doi.org/10.5194/acp-6-1201-2006, https://www.atmos-chem-phys.net/6/1201/2006/, 2006b.

Hoyle, C. R., Engel, I., Luo, B. P., Pitts, M. C., Poole, L. R., Grooß, J.-U., and Peter, T.: Heterogeneous formation of polar stratospheric clouds – Part 1: Nucleation of nitric acid trihydrate (NAT), Atmospheric Chemistry and Physics, 13, 9577–9595, https://doi.org/10.5194/acp-13-9577-2013, http://www.atmos-chem-phys.net/13/9577/2013/, 2013.

Khosrawi, F., Kirner, O., Stiller, G., Höpfner, M., Santee, M. L., Kellmann, S., and Braesicke, P.: Comparison of ECHAM5/MESSy Atmospheric Chemistry (EMAC) simulations of the Arctic winter 2009/2010 and 2010/2011 with Envisat/MIPAS and Aura/MLS observations, Atmos. Chem. Phys., 18, 8873–8892, https://doi.org/10.5194/acp-18-8873-2018, https://www.atmos-chem-phys.net/18/8873/2018/, 2018.

Spang, R., Hoffmann, L., Müller, R., Grooß, J.-U., Tritscher, I., Höpfner, M., Pitts, M., Orr, A., and Riese, M.: A climatology of polar stratospheric cloud composition between 2002 and 2012 based on MIPAS/Envisat observations, Atmos. Chem. Phys., 18, 5089–5113, https://doi.org/10.5194/acp-18-5089-2018, https://www.atmos-chem-phys.net/18/5089/2018/, 2018.

Please also note the supplement to this comment:
https://www.atmos-meas-tech-discuss.net/amt-2018-163/amt-2018-163-AC2-supplement.pdf

[Figure]

**Supplement:**

[Figure]

**Figure A5.** Examples of volume density profiles derived from CALIOP and MIPAS for observations within 200 km and 2 h during the Antarctic winter in 2011. Retrieval artifacts at 28 and 26 km altitude are visible in the MIPAS profiles (black lines) in the first column. See caption of Fig. 4 for the content of the columns.

[Figure]

**Figure A6.** Same as Fig. A5 without retrieval instabilities but with enhanced volume densities above 25 km altitude in case of both instruments.

---

## Author Comment (AC3) · 5 Oct 2018

We would like to thank H. Grothe, F. Weiss and M. J. Rossi for their helpful comments related to recent laboratory work on polar stratospheric clouds. Our answers are given below. The original comment is repeated in **bold** and changes in the manuscript text are printed in *italic*.

**Höpfner et al. present a global data set of vertical profiles of volume densities of PSCs. They have derived their data from MIPAS measurements. They use beta-NAT optical constants in the wavenumber range around 830 cm-1 in order to interpret the profiles of volume density. Strong variability of PSC parameters**

**in different Artic stratosphere winters has been observed.**

**Unfortunately, the authors ignore the fact that more than one NAT phase (i.e. alpha and beta NAT) is known from literature [1] and that both phases in combination with ice can occur in the lower polar stratosphere [2]. For both phases, optical constants have not only been calculated, but have also been measured in the laboratory with high precision [3, 4]. The morphology of the crystalline particles can have an important impact on the spectra as well [5, 6].**

We thank the team for pointing out the important issue of several NAT phases having been observed in the laboratory. We have to stress here that the goal of the presented work has been to introduce a first climatology on PSC volume densities over the whole MIPAS period. The retrieval approach is based on extensive studies by Höpfner et al., 2006b, where a variety of available optical constants have been tested on their compatibility with the MIPAS observations in combination with ground-based LIDAR measurements. In that work we have shown that a certain set of refractive indices for beta-NAT, STS and ice fitted the spectra over a broad range of wavelengths. Thus, it seems quite certain that PSCs of this composition are present in the stratosphere. This was the reason to use the optical constants related to those best-fits as baseline for the general climatology dataset. This, of course, does not mean that we generally exclude a possible existence of other phases/compositions of PSC particles. This will be pointed out more clearly in the revised version of our paper. However, since up to now those have not been shown to exist in the atmosphere, we have decided not to consider those in the present work. An extensive search for different compositions is, thus, beyond the scope of this paper.

We highly appreciate the work performed on spectroscopy in the laboratory. It is an indispensable prerequisite for meaningful analysis of remote-sensing data. However, since our work (Höpfner et al., 2006b) we are not aware of any new sets of refractive indices available for NAT or NAD and their different phases. The quoted papers do provide spectra of absorption (i.e. the imaginary part of the refractive index), however,

both, the real and imaginary part of the refractive indices are not provided. As described in our work, for simulation of the MIPAS infrared limb emission observations we have to consider also scattering, even at wavelengths around 12 $\mu$m. Therefore, we would like to take this opportunity to formulate a 'wish list' for possible future spectroscopic studies on particles to be used for analysis of remote sensing data: (1) as mentioned, the real and imaginary parts of the refractive indices are needed and, (2), generally, laboratory IR data of particles are recorded with a spectral resolution of a few wavenumbers. However, since hyperspectral remote sensing IR measurements are often much better resolved, optical constants with less than one wavenumber are requested for the analysis of very sharp aerosol bands in the IR (this related especially to the $\nu_2$-band of $NO_3^-$, which is very characteristic for NAT and NAD and which is situated in a spectral region not much affected by trace-gas interference).

**Beside NAT also NAD is a possible phase in PSCs [7ab]. Also NAD exhibits two crystalline modifications (alpha and beta NAD) [8]. Cold chamber experiments show that the metastable low temperature phase is more likely [9].**

As reported in Höpfner et al., 2006b we had performed an extensive search for the characteristic feature of NAD at 810 cm$^{-1}$ in all available MIPAS observations up to that date. However, we found no clear indication for such a spectral signature (while the NAT-signature is found frequently). For the reasons given above, we have not considered NAD in the development of the retrieval scheme. Again this will be pointed out more clearly in the revised version of our paper.

**The authors should include these latest spectroscopic and mechanistic results from the literature into their discussion. Eventually, this will help to understand the reported large variabilities.**

We agree. As suggested, we will add a paragraph in the discussion pointing to the latest laboratory work, which might be relevant for our retrieval results. However, we do not believe that the observed large variability in PSC volume density between different Arctic stratospheric winters is due to some errors in the optical constants we have utilized in our analysis. E.g. as reported by Pitts et al., 2018, CALIPSO observed very similar patterns of Arctic variability which is mainly caused by the variable meteorological stratospheric conditions between the different years.

**In addition to the points raised above it behooves the authors to compare the complex index of refraction with absorption cross section data recently published in the literature [4]. The authors should use information from laboratory experiments to calibrate, or at least validate the field observations using independent verification. Retrievals do not mean anything unless the optical data are validated using information external to the retrieval cycle. Table 6 of reference 4 displays quantitative data on optical constants of alpha- and beta-NAT as well as NAD (exact phase unknown). The associated spectra also show that all three nitric acid hydrates absorb around 830 cm-1 such that basing the assessment of the occurrence frequency solely on a single wavelength region is a losing proposition. It behooves the authors to make better use of published results from laboratory experiments in order to maximize the scientific insight (phase, frequency of occurrence, interconversion dynamics and the like) to the benefit of the readers who will appreciate a break-out from the beaten path from time to time.**

As outlined before, the intension of this work was not to perform detailed scientific investigations on particle composition, as has been performed previously. Instead we provide a first dataset over ten years, e.g. for the validation of atmospheric models, limiting the possible range of PSC vertical profiles of volume density being in accordance with the MIPAS limb radiance observations. In the selection of the spectral range, which minimizes retrieval errors, we had to balance out various side conditions, as trace gas interference, low sensitivity to scattering, and various refractive indices. With respect to the variability of refractive indices at 830 cm$^{-1}$, for the sensitivity analysis we have used refractive indices of those data which are definitively observed as composition of

PSC particles in the stratosphere (beta-NAT, STS, ice). The variability of these optical constants at 830 cm$^{-1}$ is similar to the variability of laboratory observations also for phases like alpha-NAT, beta-NAT and NAD (see e.g. Woiwode et al., 2016, Fig. 13; Ortega et al., 2006, Fig. 3).

The following text has been added in the manuscript:

*A further assumption we have made during the development of the retrieval was the choice of the optical constants. Our baseline was to use only those refractive indices of PSC composition and phase which have already been observed in the atmosphere and shown best compatibility with infrared limb observations, i.e. beta-NAT, STS, and ice (Höpfner et al., 2006b). This may lead to the following uncertainties: (1) the optical constants themselves are not perfect (Ortega et al., 2006; Iannarelli and Rossi, 2015). (2) particles may be present with different phases and composition (e.g. alpha-NAT, alpha-NAD, beta-NAD) as laboratory studies indicate their possible existence under polar stratospheric conditions (Grothe et al., 2008; Stetzer et al., 2006; Möhler et al., 2006). And (3), PSC particle shapes different from spherical ones, as seen in the laboratory (Grothe et al., 2006), can have an effect even at wavelengths in the thermal infrared (Wagner et al., 2005; Woiwode et al., 2016). We have implicitly accounted for those errors by the large variability of optical constants of beta-NAT, STS, and ice during the optimisation of the retrieval baseline configuration. Still, the use of one specific set of refractive indices leads to systematic retrieval errors which strongly depend on the atmospheric scene. A validation of infrared limb observations by in-situ measurements, especially of such cases where solid nitric acid containing particles are present, would be helpful to get a better grip on those uncertainties.*

References:

[6] Grothe, H., Tizek, H.,Waller, D., and Stokes, D. J.: The crystallization kinetics and morphology of nitric acid trihydrate, Phys. Chem. Chem. Phys., 8, 2232–2239, https://doi.org/10.1039/B601514J, http://dx.doi.org/10.1039/B601514J, 2006.

[5] Grothe, H., Tizek, H., and Ortega, I. K.: Metastable nitric acid hydrates—possible constituents of polar stratospheric clouds?, Faraday Discuss., 137, 223–234, https://doi.org/10.1039/B702343J, http://dx.doi.org/10.1039/B702343J, 2008.

Höpfner, M., Luo, B. P., Massoli, P., Cairo, F., Spang, R., Snels, M., Di Donfrancesco, G., Stiller, G., von Clarmann, T., Fischer, H., and Biermann, U.: Spectroscopic evidence for NAT, STS, and ice in MIPAS infrared limb emission measurements of polar stratospheric clouds, Atmos. Chem. Phys., 6, 1201–1219, https://doi.org/10.5194/acp-6-1201-2006, https://www.atmos-chem-phys.net/6/1201/2006/, 2006b.

[4] Iannarelli, R. and Rossi, M. J.: The mid-IR Absorption Cross Sections of alpha- and beta-NAT (HNO3-3H2O) in the range 170 to 185 K and of metastable NAD (HNO3-2H2O) in the range 172 to 182 K, J. Geophys. Res., 120, 11,707–11,727, https://doi.org/10.1002/2015JD023903, https://agupubs.onlinelibrary.wiley.com/doi/abs/10.1002/2015JD023903, 2015.

[7b] Möhler, O., Bunz, H., and Stetzer, O.: Homogeneous nucleation rates of nitric acid dihydrate (NAD) at simulated stratospheric conditions - Part II: Modelling, Atmos. Chem. Phys., 6, 3035–3047, https://doi.org/10.5194/acp-6-3035-2006, https://www.atmos-chem-phys.net/6/ 3035/2006/, 2006.

[3] Ortega, I. K., Maté, B., Moreno, M. A., Herrero, V. J., and Escribano, R.: Infrared spectra of nitric acid trihydrate (beta-NAT): A comparison of available optical constants and implication for the detection of polar stratospheric clouds (PSCs), Geophys. Res. Lett., 33, 19 816, https://doi.org/10.1029/2006GL026988, 2006.

Pitts, M. C., Poole, L. R., and Gonzalez, R.: Polar stratospheric cloud climatology based on CALIPSO spaceborne lidar measurements from 2006 to 2017, Atmos. Chem. Phys., 18, 10 881–10 913, https://doi.org/10.5194/acp-18-10881-2018, https://www.atmos-chem-phys.net/18/10881/2018/, 2018.

[7a] Stetzer, O., Möhler, O., Wagner, R., Benz, S., Saathoff, H., Bunz, H.,

and Indris, O.: Homogeneous nucleation rates of nitric acid dihydrate (NAD) at simulated stratospheric conditions - Part I: Experimental results, Atmos. Chem. Phys., 6, 3023–3033, https://doi.org/10.5194/acp-6-3023-2006, https://www.atmos-chem-phys.net/6/3023/2006/, 2006.

[1] Tizek, H., E. Knözinger, H. Grothe: "Formation and Phase Distribution of Nitric Acid Hydrates in the Mole Fraction Range xHNO3 < 0.25: a combined XRD and IR study"; Physical Chemistry Chemical Physics, 6 (2004), 972 - 979.

[8] Tizek, H., E. Knözinger, H. Grothe:"X-ray diffraction studies on nitric acid dihydrate"; Physical Chemistry Chemical Physics, 4 (2002), 5128 - 5134.

[9] Wagner, R., Möhler, O., Saathoff, H., Stetzer, O., and Schurath, U.: Infrared Spectrum of Nitric Acid Dihydrate: Influence of Particle Shape, J. Phys. Chem. A, 109, 2572–2581, https://doi.org/10.1021/jp044997u, https://doi.org/10.1021/jp044997u, pMID: 16833561, 2005.

[2] F. Weiss, F. Kubel, O. Galvez, M. Hölzel, S. F. Parker, P. Baloh, R. Iannarelli, M.J. Rossi, H. Grothe: "Metastable Nitric Acid Trihydrate in Ice Clouds"; Angewandte Chemie - International Edition, 55 (2016), 10; 3276 - 3280.

Woiwode, W., Höpfner, M., Bi, L., Pitts, M. C., Poole, L. R., Oelhaf, H., Molleker, S., Borrmann, S., Klingebiel, M., Belyaev, G., Ebersoldt, A., Griessbach, S., Grooß, J.-U., Gulde, T., Krämer, M., Maucher, G., Piesch, C., Rolf, C., Sartorius, C., Spang, R., and Orphal, J.: Spectroscopic evidence of large aspherical beta-NAT particles involved in denitrification in the December 2011 Arctic stratosphere, Atmos. Chem. Phys., 16, 9505–9532, https://doi.org/10.5194/acp-16-9505-2016, http://www.atmos-chem-phys.net/16/9505/2016/, 2016.